# UAV$^2$: A Unified and Adaptive Scheduling Framework for UAV Autopilot Systems with Reinforcement Learning

Zeying Li [1]   Shuai Zhao [1] [*]   Chaowen Wu [1]   Boyang Li [1]   Kai Huang [1]

## Abstract

Unmanned aerial vehicle (UAV) autopilot systems typically comprise navigation and flight-control modules, and their effective scheduling is critical to achieving high flight performance. However, most existing UAV platforms adopt a split architecture in which navigation and flight control are deployed on separate hardware devices. This separation restricts system-wide observability and prevents holistic scheduling and optimization across the entire autopilot pipeline. Moreover, autonomous flight performance emerges from implicit, cross-coupled, and accumulated interactions among multiple factors, rendering traditional model-based or heuristic scheduling approaches ineffective. To address these challenges, we propose UAV$^2$, a unified and adaptive scheduling framework for UAV autopilot systems with reinforcement learning, targeting flight performance optimization. UAV$^2$ integrates navigation and flight control onto a single onboard computing platform and operating system, formulates the scheduling problem as a partially observable Markov decision process, and learns scheduling policies from runtime execution feedback. The proposed approach is trained and evaluated in a hardware-in-the-loop simulation environment. Experimental results demonstrate that the learned scheduling policy consistently outperforms fixed-rate scheduling strategies in terms of flight robustness and tracking performance.

## 1. Introduction

Recent advances in artificial intelligence have significantly improved UAV autonomy, enabling rapid deployment of UAVs in complex scenarios such as transportation and large-area inspection (Zhou et al., 2020b; Menouar et al., 2017; Jordan et al., 2018), etc. A typical UAV autopilot pipeline mainly consists of the navigation module for decision making (e.g., perception, localization, planning (Chiang et al., 2017; Zhu et al., 2025; Tordesillas et al., 2019)) and flight control module for execution (e.g., state estimation and control loop (Meier et al., 2015; Baldi et al., 2022)), deployed on embedded platforms with stringent size, weight, and power constraints. Because control and navigation are tightly coupled and require frequent information exchange, UAV flight performance is determined not only by the control and navigation algorithms, but also by scheduling and allocation of these tasks under limited hardware resources that govern their timely interaction.

However, despite the strong interactions between the two modules, most existing autonomous UAV platforms are built upon a split hardware architecture that separates navigation and flight control, where the navigation module is deployed on a companion computer while the control module operates on a dedicated flight control unit (FCU) (Ma et al., 2020; Zhou et al., 2019; Qin et al., 2018; Bigazzi et al., 2021). This causes the autopilot taskset to span across different clocks, OS kernels and hardware. As a result, the scheduling is restricted to the companion computer, leaving the FCU as a black box and preventing the holistic design of the system, thereby hindering effective optimizations of the flight performance.

In addition, UAV system scheduling poses a fundamental modeling challenge, since the impact of scheduling decisions on flight performance is mediated by tightly coupled and implicit interactions across navigation, flight control, environmental dynamics, and computational resources (Demirel et al., 2018; Glaubius et al., 2010). Moreover, scheduling actions rarely align with instantaneous performance metrics. Their effects propagate through the autopilot pipeline and are often apparent after significant delays, e.g., short-term throttling of localization may only surface later as position drift. As a result, constructing accurate quantitative models for these partially unobservable effects is difficult (Chen et al., 2021; Xiang & Foo, 2021; Kurniawati, 2022; Lauri et al., 2022), limiting the applicabil-

---

[1]School of Computer Science and Engineering, Sun Yat-sen University, Guangzhou, China. Correspondence to: Shuai Zhao <zhaosh56@mail.sysu.edu.cn>.

*Proceedings of the 43$^{rd}$ International Conference on Machine Learning*, Seoul, South Korea. PMLR 306, 2026. Copyright 2026 by the author(s).

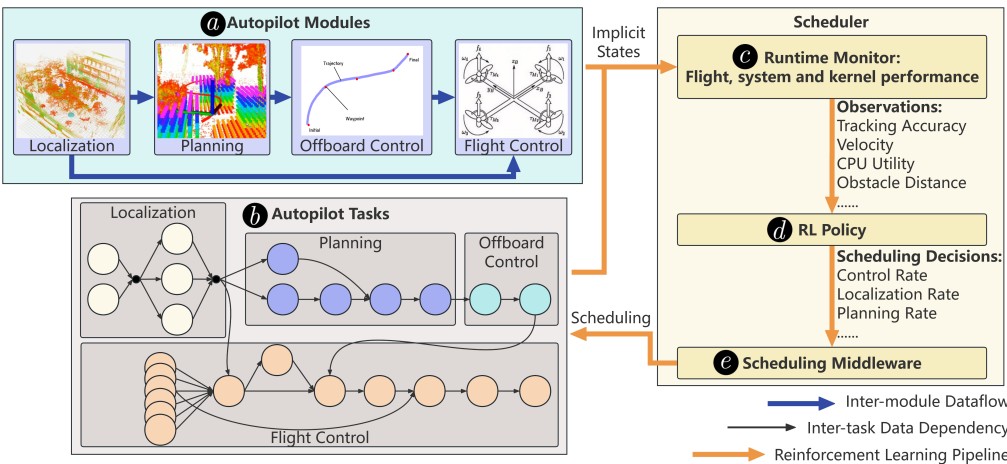

*Figure 1.* **Overall system architecture of the unified scheduling framework.** The UAV autopilot system integrates localization, planning, offboard control, and flight control modules on a unified computing platform (*a*), where modules are decomposed into schedulable tasks (*b*). A runtime monitor (*c*) provides system and flight observations to a reinforcement learning policy (*d*), and the resulting scheduling decisions are enforced through a unified scheduling middleware (*e*), ultimately leading to improved flight performance.

ity of many analytical and heuristic scheduling approaches.

**Contribution.** To address the above limitations, this work presents UAV$^2$, a unified and adaptive scheduling framework for UAV autopilot systems based on reinforcement learning, targeting flight performance optimization[1]. The contributions are summarized as follows:

1. We propose a unified UAV system architecture that integrates navigation and flight control into a single computational platform, enabling system-wide observability and holistic scheduling of the UAV system.

2. We formulate UAV system scheduling as a partially observable Markov decision process (POMDP), capturing the implicit, delayed, and cross-coupled effects of scheduling decisions on flight performance.

3. Based on this formulation, we design a reinforcement learning-based runtime scheduler that learns scheduling policies from online execution feedback, and demonstrate consistent improvements in flight robustness and trajectory tracking over fixed-rate scheduling baselines in a hardware-in-the-loop simulation framework with real embedded UAV hardware.

## 2. Framework Design

This section introduces the overall system architecture of UAV$^2$, as illustrated in Fig. 1. Navigation and flight-control modules execute on a unified onboard platform (Fig. 1*a*) and are exposed as schedulable execution units (Fig. 1*b*). Runtime observations are collected by a monitor (Fig. 1*c*) and provided to an RL policy (Fig. 1*d*), whose scheduling actions are enforced through dedicated middleware

---

[1]An example implementation of the proposed UAV$^2$ framework is available at https://github.com/leeziy/UAV-2-RL

(Fig. 1*e*).

This section does not present the learning algorithm itself, but establishes the system abstractions required by the learning problem: how execution units are exposed for control, how observations are constructed, and how decisions are applied across the unified system. The reinforcement learning formulation and policy design are presented in Section 3.

The remainder of this section details the unified scheduling infrastructure, the runtime monitor, and the scheduling middleware.

### 2.1. Unified Scheduling Infrastructure

UAV$^2$ is built upon a unified UAV system architecture in which navigation and flight-control tasks are integrated onto a single onboard computing platform and operating system. This unified design is a prerequisite for the proposed RL-based runtime scheduling, as it enables coherent observation of system-wide execution signals and consistent application of scheduling decisions across the UAV system.

At the application layer, the UAV system runs a complete autopilot pipeline (see Fig. 1*a*). We adopt FAST-LIO2 (Xu & Zhang, 2021; Xu et al., 2022) for localization, EGO-Planner (Zhou et al., 2020a; 2021) for trajectory planning, and PX4-Autopilot (Meier et al., 2015) for flight control. We also add an offboard control module that translates planned trajectories into waypoint commands. A detailed description of the dataflow and inter-module dependencies is provided in Appendix A. All modules are adapted to operate within the unified architecture and to expose task-level execution controls required by the scheduling framework.

At the kernel layer, the proposed framework exposes the

autopilot modules as a set of independently schedulable execution units (see Fig. 1❷). This task-level abstraction allows scheduling decisions to be realized as controlled adjustments to execution behavior, while leaving low-level dispatching to the operating system. Due to different execution models across autopilot components, this abstraction cannot be achieved directly.

PX4-Autopilot manages tasks through the WorkQueue mechanism: tasks with the same priority share a FIFO WorkQueue, and each WorkQueue is executed by a single NuttX thread or POSIX thread (Meier et al., 2015). We modify this binding by assigning each task to a private WorkQueue backed by a dedicated thread, allowing individual management of each flight control task. As for the navigation modules (i.e., FAST-LIO2, EGO-Planner, Offboard Control), they are all based on the ROS1 environment and infrastructure. In ROS1, a `Callback` triggered by a `Timer` or a `Topic` is enqueued into a `CallbackQueue`, and ready callbacks are executed by worker threads managed by `spin` or `AsyncSpinner` in FIFO order (Quigley et al., 2009; Saito et al., 2018). Therefore, we associate each task with a dedicated `Callback`, `NodeHandle`, `CallbackQueue`, and `AsyncSpinner`, ensuring that each task executes within an individual thread and can be explicitly triggered via a `Topic` message.

Through these modifications, components with different execution models are uniformly represented as schedulable threads under the operating system, forming the basis for the scheduling middleware and enabling system-level reasoning about scheduling decisions.

## 2.2. Runtime Monitoring

The runtime monitor provides the observation interface by continuously collecting execution-related signals from the running UAV system (see Fig. 1❸). These signals include flight-level feedback, system behavior, and kernel-level execution information for RL-based scheduling.

To reduce implementation complexity, the monitor is encapsulated as a ROS1 node and integrated into the native ROS1 communication and execution infrastructure (see Fig. 2). In practical deployments, the RL policy and ROS may reside in different system environments, and training and online inference are often conducted on separate devices. Monitor outputs are transmitted through UDP sockets, decoupling runtime monitoring from the execution platform and allowing training, evaluation, and deployment to run in different system environments.

The runtime monitor is purely observational and does not perform scheduling or modify task execution. All scheduling decisions are generated by the RL policy described in Section 3, and are applied exclusively through the schedul-

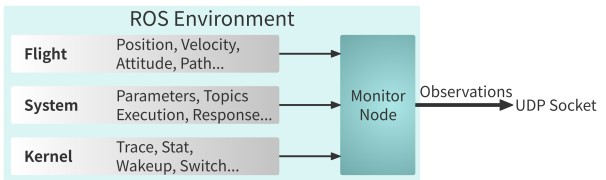

*Figure 2.* Basic structure of the monitor.

ing middleware described in the following subsection.

## 2.3. Scheduling Middleware

Overall, the system consists of 25 threads distributed across modules as scheduling targets. Directly controlling all individual threads would lead to an excessively large and intractable action space. Instead, tasks with tightly coupled functional roles and data dependencies are grouped as unified scheduling units, reducing action dimensionality while preserving the main trade-offs that govern flight performance. These threads are organized into four scheduling groups: 13 flight-control threads, 5 localization threads, 5 trajectory-planning threads, and 2 offboard control threads (see Table 5 in Appendix B for details).

As an engineering safeguard for real-time execution and system stability, the RL policy operates through an abstracted action space rather than directly controlling worker threads. The policy is deliberately restricted to bounded frequency-level decisions and does not modify thread priorities or scheduling policies, thereby preserving kernel-level scheduling invariants during policy exploration and deployment. An intermediate scheduling layer is introduced to map and execute the discrete action decisions produced by the RL policy (see Fig. 1❹).

The scheduling middleware statically pins the flight-control threads to the highest priority under SCHED_FIFO, while delegating their wake-up and dispatching to the OS kernel. It exposes only a frequency-setting interface, which configures the execution rate based on action decisions from RL policy.

The navigation threads are scheduled by a lightweight dispatcher. At each scheduling tick, the dispatcher determines whether to wake a specific thread group based on the action decision output by the RL policy. Within each tick, the wake-up time of individual threads is further adjusted by delaying the transmission of trigger messages. Accordingly, the wake-up order and timing of the threads can be predefined based on task data dependencies and their execution times.

From a system perspective, this middleware provides a deterministic mapping from scheduling decisions to concrete execution timing patterns across task groups.

# 3. Reinforcement Learning Design

Building on the unified system architecture described in Section 2, this section formulates UAV system scheduling as a reinforcement learning problem. As discussed in Section 1, scheduling decisions influence flight performance through complex and delayed interactions among navigation, estimation, control, and computational resources. These interactions give rise to latent system dynamics that are only partially observable through runtime measurements, making it difficult to construct accurate analytical models or effective heuristic scheduling rules based on instantaneous metrics or explicit dynamics models.

We therefore formulate scheduling as a partially observable Markov decision process (POMDP) (Cassandra, 1998; Littman, 2009; Spaan, 2012), and design the observations, actions, and rewards accordingly. An RL policy then learns implicit system dynamics and the long-term consequences of scheduling decisions from execution feedback.

In this formulation, the latent state is the joint physical and computational state of the unified UAV system, including the vehicle motion state, sensor streams and buffers, internal states of localization, planning, and control modules, OS-level execution states, resource states, and the current scheduling configuration. The runtime monitor observes only a subset of these variables, and transitions are induced jointly by UAV dynamics, sensor generation, autopilot computation, and OS scheduling behavior. Therefore, the policy operates on observation and action histories rather than on a fully observed system state. Appendix C provides the complete POMDP definition.

## 3.1. Observation Design for Scheduling

The observation design supports policy learning for unified UAV scheduling, where scheduling decisions induce delayed and nonlinear effects on flight performance. Because instantaneous measurements are insufficient under partial observability, observations integrate current runtime signals with historical context, aiding the policy in inferring long-term scheduling effects.

As summarized in Table 1, the observation combines system-level resource indicators, scheduling configurations, flight-

*Table 1.* Observations used for RL-based scheduling

| Category | Observation Variable |
|---|---|
| System | CPU time consumed by flight control tasks |
| System | CPU time consumed by navigation tasks |
| Schedule | Frequency of flight control tasks |
| Schedule | Frequencies of navigation task groups |
| Flight | Deviation from the planned trajectory |
| Flight | Current vehicle velocity |
| Environment | Distance to the nearest obstacle |

level feedback, and environmental context, and is constructed by the runtime monitor (see Section 2.2) at each RL decision step. System-related signals measure CPU time consumed by flight control and navigation tasks, providing a direct measure of resource pressure and cross-task interference, while scheduling-related signals encode the current execution frequencies.

To handle delayed and cumulative effects, observations are provided as a fixed-length sliding-window buffer with a history length of 10. Historical scheduling actions are included in the same temporal window, so runtime signals and action information are retained together. This temporal stacking allows the policy to condition decisions on recent system states and scheduling configurations while keeping the observation dimensionality bounded.

## 3.2. Action Space and Scheduling Decisions

At each decision step, the policy selects a scheduling configuration that specifies the execution rates of a set of schedulable tasks in the unified UAV system. Concretely, each action corresponds to a discrete combination of task-level rate settings, such as the update frequencies assigned to localization, planning, and control modules managed by the scheduling middleware described in Section 2.

As shown in Table 2, the action space consists of two components. The first component selects the execution frequency of flight-control tasks from a discrete set of $\{100, 200, 300, 400\}$ Hz. All flight-control threads are statically assigned the highest priority and scheduled by the kernel under the SCHED_FIFO policy. The policy does not modify thread priorities or scheduling policies. Instead, it regulates the activation rate of flight-control tasks by selecting among predefined frequency levels, thereby adjusting the control-loop update rate and the associated computational load.

The second component determines the group-level execution frequencies of navigation tasks. Ten navigation threads are organized into three functional groups based on data dependencies and execution roles. These threads are not directly scheduled by the policy at the kernel level. Instead, they are activated by a lightweight user-space dispatcher, which triggers task execution at group-level frequencies selected by the policy from a discrete set of $\{10, 20, 30, 40, 50\}$ Hz. This activation-based mechanism allows the policy to influ-

*Table 2.* Action space for RL-based UAV scheduling

| Target Tasks | Discrete Frequencies |
|---|---|
| Flight Control | $\{100, 200, 300, 400\}$ Hz |
| Localization | $\{10, 20, 30, 40, 50\}$ Hz |
| Planning | $\{10, 20, 30, 40, 50\}$ Hz |
| Offboard Control | $\{10, 20, 30, 40, 50\}$ Hz |

ence execution timing through rate selection, without direct manipulation of kernel scheduling behavior.

The frequency sets are chosen from feasible operating ranges of the corresponding modules, considering real-time requirements, measured execution costs, and persistent-overload avoidance. This structured discrete action design keeps the action space tractable for stable and sample-efficient learning under partial observability, while restricting decisions to task-group frequency configurations that directly allocate computational resources and affect flight performance, as defined in Section 2.3 and summarized in Table 5 in the Appendix.

### 3.3. Reward Function Design

The reward function guides scheduling decisions toward improved long-term flight outcomes while respecting computational feasibility and safety. It scalarizes flight behavior, resource feasibility, and collision avoidance, and is formulated entirely in terms of penalties. Formally, the reward at time step $t$ is defined as

$$
\begin{aligned}
r_t = & -\lambda_{\text{traj}}\, d_{\text{traj}}\big(\mathcal{P}_t^{\text{odom}}, \mathcal{P}_t^{\text{ref}}\big) \\
& -\lambda_{\text{est}}\, d_{\text{traj}}\big(\mathcal{P}_t^{\text{odom}}, \mathcal{P}_t^{\text{gt}}\big) \\
& -\lambda_{\text{cpu}}\, \mathbb{I}(u_t > 0.95) \\
& -\lambda_{\text{col}}\, \mathbb{I}(\text{collision}_t)\,.
\end{aligned}
\tag{1}
$$

where $\mathcal{P}_t^{\text{odom}}$, $\mathcal{P}_t^{\text{ref}}$, and $\mathcal{P}_t^{\text{gt}}$ denote the odometry, reference, and ground-truth trajectory segments within a fixed temporal window ending at time $t$. The function $d_{\text{traj}}()$ computes the bidirectional minimum average distance between two trajectory segments, serving as a windowed measure of trajectory deviation rather than a pointwise error.

As summarized in Table 3, the performance-related penalties capture both trajectory-tracking accuracy and state estimation quality. The two flight-performance terms, measuring deviation from the planned trajectory and from ground truth, are assigned identical weights ($\lambda_{\text{traj}} = \lambda_{\text{est}} = 0.01$). The deviation between odometry and the planned trajectory reflects flight behavior, while the deviation from ground truth penalizes degraded onboard localization. These terms operate continuously over time and their raw values typically remain bounded. Setting their weights at the same order of magnitude encourages the policy to balance tracking accuracy and estimation quality without allowing either to

*Table 3.* Reward components used for RL-based scheduling

| Category | Reward Term | Weight |
|---|---|---|
| Flight | Deviation from planned trajectory | $\lambda_{\text{traj}} = 0.01$ |
| Flight | Deviation from ground truth | $\lambda_{\text{est}} = 0.01$ |
| System | CPU utilization $> 95\%$ | $\lambda_{\text{cpu}} = 0.1$ |
| Safety | Collision (terminate) | $\lambda_{\text{col}} = 1$ |

dominate the learning signal.

To discourage scheduling behaviors that compromise computational resource feasibility, a system-level penalty is applied when total CPU utilization exceeds 95%. The penalty uses a medium weight ($\lambda_{\text{cpu}} = 0.1$), reflecting its role as a soft feasibility constraint rather than a primary performance objective. Since overload events are sparse but destabilizing when sustained, this weight ensures that prolonged saturation produces a noticeable cumulative penalty while permitting brief overloads when required to preserve flight stability.

Collision handling is incorporated during training through a terminal penalty. The collision penalty is assigned the largest weight ($\lambda_{\text{col}} = 1$), one to two orders of magnitude higher than other terms. This scaling makes collision events dominate the reward signal and discourages the policy from trading collision risk against incremental performance or efficiency gains. This mechanism is a training-time safeguard rather than a formal safety guarantee.

Overall, the reward weights are designed to balance flight performance, computational resources, and safety, enabling long-horizon credit assignment under delayed and nonlinear scheduling effects.

### 3.4. RL Policy

As discussed above, the UAV system scheduling problem is formulated as a POMDP. We therefore employ an LSTM-augmented PPO policy (Recurrent PPO), which maintains a latent memory state over observation histories to address partial observability and delayed effects (Schulman et al., 2017; Duoxiu et al., 2022; Yau et al., 2024; Pleines et al., 2022; Huang & Qu, 2023).

The primary contribution lies in the problem formulation rather than proposing a new learning algorithm. Given the structured observation, action, and reward formulation described above, the policy approximates a history-dependent scheduling strategy from execution feedback. Standard recurrent actor-critic methods are sufficient in this setting, and alternative recurrent reinforcement learning algorithms are expected to yield comparable behavior under the same formulation.

## 4. Training Framework

### 4.1. Hardware-in-the-Loop Simulation Framework

The proposed scheduling policy is trained using a hardware-in-the-loop (HIL) simulation framework that tightly couples a high-fidelity physical simulator with a real onboard computing platform (Fig. 3). In this framework, physical dynamics and sensor measurements are simulated using Gazebo Classic, while the complete UAV autopilot taskset

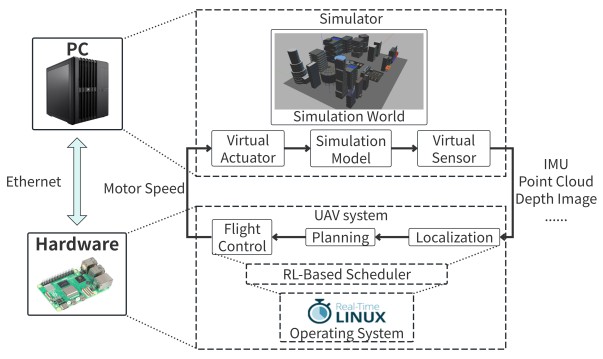

*Figure 3.* Architecture of the hardware-in-the-loop simulation framework.

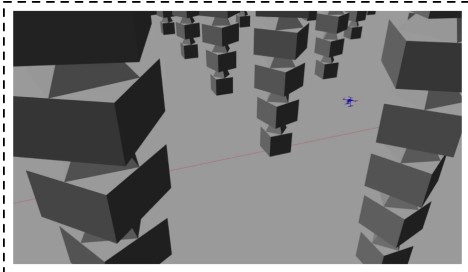

*Figure 4.* Experimental scenario used in HIL simulations.

executes on a real embedded onboard computer. Unlike purely software-based simulations, this HIL setup allows scheduling decisions to directly affect the execution of real navigation and control tasks under a real operating system, thereby exposing the policy to realistic flight dynamics, stochastic sensor noise, and dynamic OS scheduling behavior. As a result, the learning signals exhibit statistical characteristics that are more representative of those encountered during policy execution. A more detailed description of the HIL simulation framework is provided in Appendix F.

The simulator runs on a host workstation equipped with an AMD Ryzen$^{\text{TM}}$ 9 9700X CPU featuring eight physical cores with a maximum frequency of 5.5 GHz. The host operating system is Ubuntu 20.04.6 LTS x86_64 with Linux 6.18.2.

The onboard computing platform adopts a Rockchip RK3588S SoC, which integrates 4 Cortex-A55 and 4 Cortex-A76 cores. It runs Ubuntu 20.04 with Linux 5.10.198 PREEMPT_RT. To enable fine-grained runtime analysis, `Trace`-related kernel configurations are enabled to capture execution events such as `sys_call`, and `SCHEDSTATS` is enabled to provide high-accuracy CPU time measurements. Computational resources are partitioned using cgroups. The 4 Cortex-A55 cores and 2 Cortex-A76 cores are allocated to system daemons, background services and kernel threads, while the remaining 2 Cortex-A76 cores exclusively execute the 25 worker threads of the autopilot taskset and serve as the scheduling space.

The onboard hardware and the simulator are connected via wired Ethernet with a bandwidth of 1 Gb/s and a round-trip latency below 1 ms. This latency is significantly shorter than the periods of all sensors and control tasks, and therefore does not introduce measurable interference to the system behavior.

All simulations are conducted in the Pillars environment shown in Fig. 4. The dense obstacle layout increases sensitivity to estimation and planning quality under constrained computational resources, making it particularly suitable for

training and evaluating scheduling strategies.

### 4.2. Training Setup

The reinforcement learning policy interacts with the system at a fixed decision frequency of 10 Hz. System observations are constructed from sliding-window statistics as described in Section 3.1. Scheduling actions select discrete execution frequencies and are sent to the scheduling middleware via a UDP socket. Full training hyperparameters and policy architecture details are provided in Appendix E, and the corresponding implementation and automation scripts are available in the code repository linked in Section 1.

Training is conducted in an episodic manner, where each episode corresponds to a complete autonomous flight run. Collisions are detected directly from physics engine contact feedback, and mission completion is signaled by the navigation module, both of which trigger episode termination.

### 4.3. Training Dynamics

Fig. 5 summarizes the training dynamics of the proposed scheduler in the hardware-in-the-loop setting. Learning signals are inherently noisy due to stochastic flight perturbations as well as the indirect and time-delayed influence of scheduling decisions on state estimation and control pipelines. To expose the underlying training trends, we report both raw measurements and exponentially smoothed curves, emphasizing long-horizon learning behavior rather than short-term fluctuations.

Fig. 5a illustrates the evolution of the trajectory deviation term used in the reward formulation. While instantaneous deviations remain highly variable throughout training, the smoothed curve exhibits a gradual downward trend. This behavior suggests that, over time, the learned policy tends to favor scheduling configurations associated with improved tracking performance under identical environmental conditions.

Fig. 5b and Fig. 5c report the episodic return and the explained variance of the value function, respectively. The episodic return initially decreases during early exploration and subsequently stabilizes, which is consistent with the

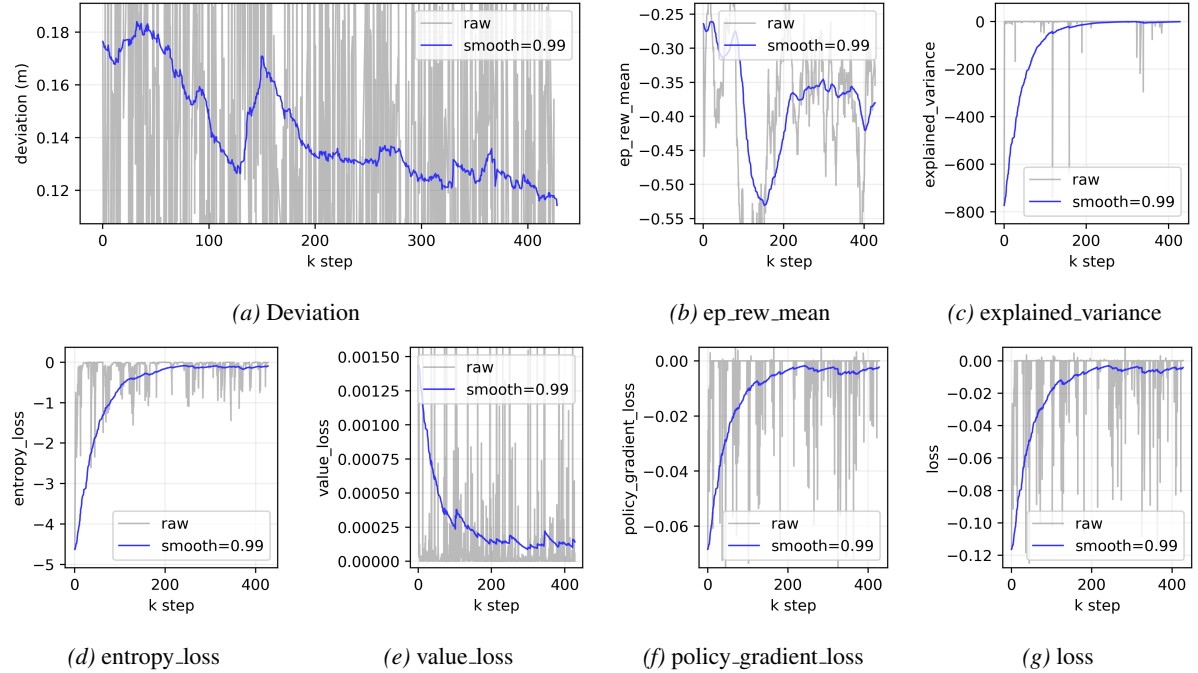

*(a)* Deviation      *(b)* ep_rew_mean      *(c)* explained_variance

*(d)* entropy_loss     *(e)* value_loss     *(f)* policy_gradient_loss     *(g)* loss

*Figure 5.* Training Record.

policy transitioning from broad exploration toward more consistent scheduling behaviors. The explained variance increases from strongly negative values and approaches zero, suggesting a reduction in inconsistency between predicted values and observed returns. However, this metric does not imply accurate modeling of system dynamics, as the critic remains fundamentally constrained by partial observability and delayed credit assignment inherent to the scheduling problem.

Optimization diagnostics are shown in Fig. 5d-5g. The entropy term gradually decreases over training, indicating reduced policy stochasticity without an abrupt collapse of exploration. Both the value loss and the policy gradient loss decrease rapidly in early training and remain bounded thereafter, suggesting stable optimization behavior with the recurrent policy architecture.

Overall, the training curves are consistent with stable learning in a partially observable scheduling environment. They do not imply explicit modeling of the underlying system dynamics, but support that the policy can improve end-to-end flight outcomes through runtime execution feedback.

## 5. Experiments and Results

To ensure a fair and consistent evaluation, the evaluation is conducted on the same HIL simulation framework and configurations as training. This setup isolates the effect of scheduling policies and allows performance differences to be attributed solely to scheduling decisions rather than changes in the execution environment.

### 5.1. Baselines and Metrics

We compare $UAV^2$ with 4 fixed-rate baselines, reflecting a static-frequency scheduling strategy commonly used in real-world UAV autopilot deployments. Each baseline sets the flight-control loop to one of $\{100, 200, 300, 400\}$ Hz. The navigation groups (localization, planning, and offboard control modules) operate at matched rates $\{20, 30, 40, 50\}$ Hz, yielding configurations (A) 100-20-20-20, (B) 200-30-30-30, (C) 300-40-40-40, and (D) 400-50-50-50.

All methods are evaluated using the same map, identical start and goal locations and mission settings. Each method is executed for 50 independent episodes under these fixed conditions.

We report four metrics to characterize system and flight performance.

1. **Mean Trajectory Tracking Deviation.** The mean deviation distance between the odometry and the planned trajectory over each episode.
2. **Success Rate.** The percentage of episodes that successfully complete the task without collision.
3. **CPU Time.** The mean CPU time used by the autopilot taskset over each episode.
4. **Flight Duration.** The total number of steps required to complete a successful episode.

### 5.2. Results and Analysis

Fig. 6 compares the learned scheduling policy with 4 fixed-rate baselines under identical test conditions.

Fig. 6a reports the episode-level mean trajectory tracking deviation. Although the highest-rate baseline (Strategy D) achieves the smallest median deviation, it exhibits substantially larger dispersion and more outliers, indicating less consistent tracking performance across episodes. In contrast, the learned policy maintains comparably low deviation while significantly reducing variance. This behavior suggests that the policy does not rely on uniformly increasing task execution rates, but instead adapts scheduling decisions to system state and flight context, resulting in more stable tracking performance.

The success rate comparison in Fig. 6b reveals a clearer advantage of the proposed approach. The learned policy achieves the highest success rate (86.0%), outperforming all fixed-rate baselines. Notably, strategies with similar median deviation still exhibit markedly lower success rates, indicating that trajectory accuracy alone does not fully capture mission robustness. This result is consistent with adaptive scheduling mitigating failure cases arising from delayed estimation errors, transient computational overloads, or unfavorable task interactions.

Fig. 6c shows the mean CPU usage across different strategies. As expected, the lowest-rate baseline minimizes computational load but suffers from degraded tracking performance and frequent mission failures. The learned policy operates at a moderate usage level, remaining well below saturation while achieving substantially improved flight outcomes. Compared with higher-rate baselines, it attains higher success rates with comparable or lower computa-

tional cost, indicating a more effective use of limited onboard computational resources.

Overall, these results are consistent with the effectiveness of the proposed reinforcement learning-based scheduler in balancing tracking accuracy, robustness, and computational efficiency. Rather than optimizing a single metric in isolation, the learned policy leverages runtime feedback to navigate the implicit and delayed coupling between scheduling decisions and flight behavior, leading to improved end-to-end performance relative to static scheduling strategies. Appendices G and H report cross-scenario results and qualitative failure analysis.

### 5.3. Reward Ablation and Sensitivity Analysis

We further evaluate the effect of reward design through the ablation study summarized in Table 4. All variants are trained for 1000 episodes and evaluated over 50 test episodes. Group A scales all reward weights jointly while preserving their relative ratios, Group B removes one reward component at a time, and Group C reassigns the same weight magnitudes across terms. Fig. 7 shows that the policy is more sensitive to relative weighting than to absolute reward scale. A1 and A2 retain success rates of 84% and 86%, respectively, compared with 86% for the default policy, and their mean deviation and CPU time remain close to the original values.

The component-wise removals clarify the role of each term. Removing the two deviation-related penalties in B1 eliminates the dense step-wise learning signal, reducing success to 42% and increasing mean deviation from 0.15 m to 0.35 m. Removing the CPU penalty in B2 raises mean CPU time from 73.99 ms to 83.15 ms, reduces success to 66%, and increases mean deviation to 0.21 m, indicating that the CPU term acts as a resource-aware regularizer. Removing the collision penalty in B3 leads to only 28% success because early collision can shorten the episode and reduce accumulated dense penalties. Group C further shows that collision avoidance must be weighted sufficiently higher than the dense deviation penalties accumulated over a successful episode. Once this balance is disrupted, success

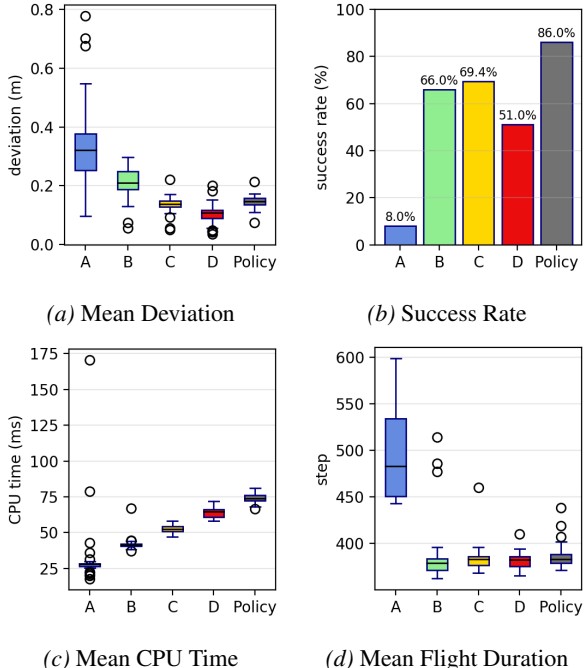

*(a)* Mean Deviation      *(b)* Success Rate

*(c)* Mean CPU Time      *(d)* Mean Flight Duration

*Figure 6.* Test results under different scheduling strategies.

*Table 4.* Reward ablation and sensitivity study.

| Setting | Purpose | $\lambda_{\text{traj}}$ and $\lambda_{\text{est}}$ | $\lambda_{\text{cpu}}$ | $\lambda_{\text{col}}$ |
|---------|---------|-------------|-------|-------|
| Origin | Default reward | 0.01 | 0.1 | 1 |
| A1 | Global scaling by 0.1 | 0.001 | 0.01 | 0.1 |
| A2 | Global scaling by 10 | 0.1 | 1 | 10 |
| B1 | Remove deviation penalties | 0 | 0.1 | 1 |
| B2 | Remove CPU penalty | 0.01 | 0 | 1 |
| B3 | Remove collision penalty | 0.01 | 0.1 | 0 |
| C1 | Reassign weights | 0.01 | 1 | 0.1 |
| C2 | Reassign weights | 0.1 | 0.01 | 1 |
| C3 | Reassign weights | 0.1 | 1 | 0.01 |
| C4 | Reassign weights | 1 | 0.01 | 0.1 |
| C5 | Reassign weights | 1 | 0.1 | 0.01 |

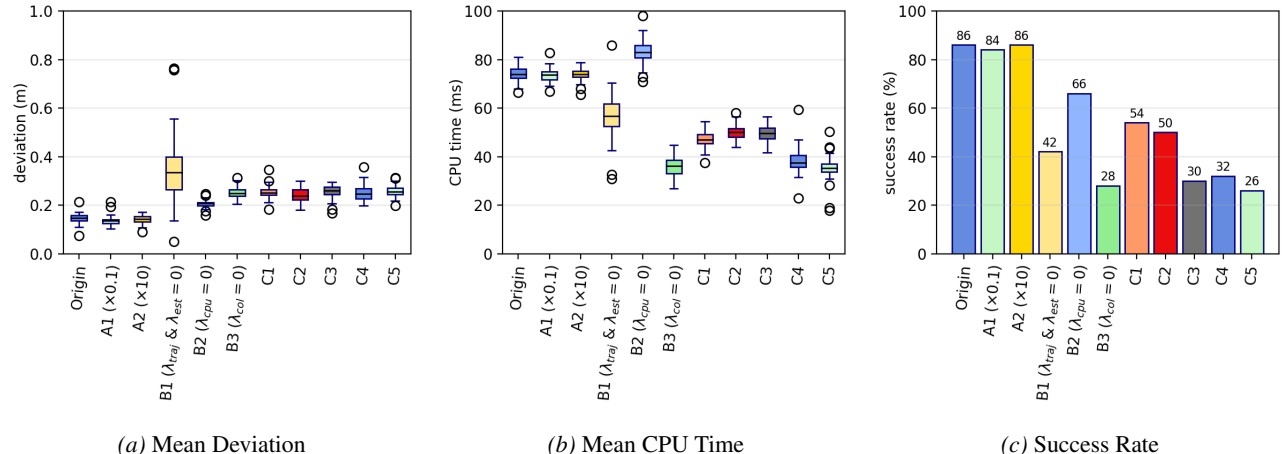

*(a)* Mean Deviation      *(b)* Mean CPU Time      *(c)* Success Rate

*Figure 7.* Reward ablation and sensitivity results.

drops to $54\%$, $50\%$, $30\%$, $32\%$, and $26\%$ for C1–C5, respectively.

### 5.4. Real World Experiment

In addition to simulation-based evaluation, we deploy the proposed scheduling framework on a real UAV platform and conduct preliminary flight experiments in a sparse forest scenario. The detailed design and implementation of the UAV is provided in Appendix J. Fig. 8 illustrates a representative flight experiment and the corresponding navigation result in ROS. The UAV uses the same onboard autonomy stack as in the HIL experiments and flies from $(0, 0, 2)$ m to $(45, 12, 2)$ m with a maximum speed of $2$ m/s.

We evaluate the learned policy and the four fixed-rate baselines using 10 real-flight trials for each strategy. As shown in Fig. 9, the learned policy achieves the highest success rate of $100\%$, compared with $40\%$, $80\%$, $90\%$, and $70\%$ for strategies A–D, respectively. Its mean trajectory tracking deviation is $0.17$ m, which is close to the high-rate baselines C and D ($0.16$ m and $0.15$ m), while avoiding the lower robustness observed in strategy D. The policy also uses higher CPU time in this real-world setting, indicating that the learned scheduler tends to allocate additional computation when operating under real sensing and environmental uncertainty. Overall, these results support the feasibility of deploying the learned scheduler on a real UAV platform, while we regard them as preliminary real-world evidence rather than a complete safety validation.

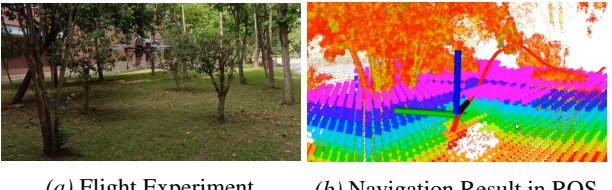

*(a)* Flight Experiment      *(b)* Navigation Result in ROS

*Figure 8.* Experimental Flight in Real World.

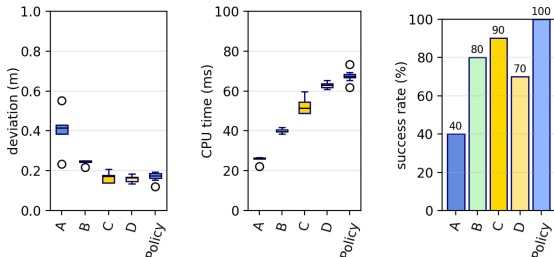

*(a)* Mean Deviation *(b)* Mean CPU Time    *(c)* Success Rate

*Figure 9.* Test results in real-world scenarios.

### 6. Conclusion

We presented UAV$^2$, a unified and adaptive scheduling framework for UAV autopilot systems with reinforcement learning. It integrates the entire autopilot taskset on a single onboard computing platform. By formulating system scheduling as a sequential decision problem under partial observability, UAV$^2$ learns to adapt task execution rates from runtime signals and flight feedback. We trained and evaluated UAV$^2$ in a hardware-in-the-loop simulation that runs the autopilot taskset on real UAV hardware while simulating high-fidelity dynamics and sensors. Across challenging obstacle-rich missions, UAV$^2$ consistently outperformed fixed-rate baselines, improving trajectory tracking while achieving a higher mission success rate. These results indicate that learning scheduling policies from real execution feedback provides a practical and effective approach toward holistic, performance-aware scheduling in resource-constrained UAV autopilot systems. A preliminary discussion of cross-hardware transfer is provided in Appendix I.

## Impact Statement

This paper presents work whose goal is to advance the field of machine learning. There are many potential societal consequences of our work, none of which we feel must be specifically highlighted here.

## Acknowledgements

This work is supported in part by National Key Research and Development Program under Grant 2023YFB4503700, National Natural Science Foundation of China under Grant 62302533, Guangdong Basic and Applied Basic Research Foundation under Grant 2024A1515010240.

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

# A. UAV Autopilot Pipeline

Here we outline the high-level dataflow of the UAV autopilot taskset (see Table 5), focusing on the major information pathways and inter-module dependencies rather than implementation details.

The autopilot pipeline integrates localization (Xu & Zhang, 2021; Xu et al., 2022), planning (Zhou et al., 2020a; 2021), and flight control (Meier et al., 2015) into a unified execution graph (see Fig. 1 ⓐ and ⓑ) running on a single onboard computing platform. These various data rates and feedback dependencies impose non-trivial scheduling constraints across the pipeline. An overview of the dataflow is illustrated in Fig. 10.

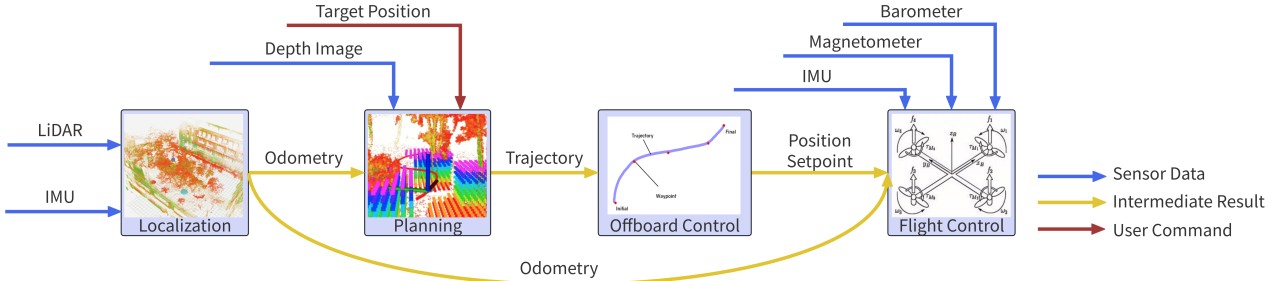

*Figure 10.* Dataflow and major inter-module dependencies in the UAV autopilot pipeline.

Raw sensor measurements, including LiDAR, IMU, depth images, and onboard proprioceptive sensors, enter the pipeline at heterogeneous rates. LiDAR and IMU streams are tightly fused to produce high-frequency odometry estimates, which form the geometric and kinematic backbone of the navigation subsystem. In parallel, depth observations are temporally aligned with odometry and used to incrementally maintain a local representation of surrounding obstacles within the planning horizon.

The navigation subsystem consumes two primary inputs: the current motion state of the UAV and a local environment representation. The motion state, comprising position, orientation, and velocity, is derived from LiDAR–inertial odometry, while the environment representation is updated using synchronized depth and pose information.

Based on these inputs, the local planner generates dynamically feasible trajectories in a receding-horizon manner. Trajectory generation and collision checking form a closed loop, where candidate trajectories are continuously validated against the local obstacle representation, and replanning is triggered when safety constraints are violated. The output of the navigation subsystem is a time-parameterized motion reference, expressed as a sequence of position setpoints or waypoints.

The interface between navigation and flight control is intentionally narrow. Navigation outputs only high-level motion references, while all low-level stabilization and actuator-level decisions are handled by the control subsystem. This design allows the planner to operate at a lower rate and tolerate higher latency, while the control loop maintains high-frequency execution to ensure stability.

The flight-control subsystem fuses inertial, magnetic, and barometric measurements with externally provided odometry to estimate the full vehicle state. This state estimate serves as the input to a cascaded control architecture consisting of position, attitude, and angular-rate control loops. High-level position setpoints are tracked by the position controller, which generates intermediate attitude and thrust commands that are further refined and allocated to individual actuators.

Although navigation and flight control are logically separated, they remain tightly coupled through shared state estimates and command interfaces. Navigation relies on timely odometry feedback to maintain planning consistency, while flight control depends on smooth and timely motion references to preserve closed-loop stability.

## B. Scheduling Targets for Middleware and Policy

Table 5 lists the major software threads of the unified UAV system that are exposed as schedulable targets. These threads are grouped according to their functional roles in the autonomy stack, including flight control, LiDAR–inertial odometry, path planning, and autonomous flight control.

For each thread, the table reports its functionality and the measured worst-case execution time (WCET) on the target onboard platform. The flight control thread group consists of high-frequency sensing, state estimation, and closed-loop control tasks with relatively small WCETs, reflecting their real-time criticality. In contrast, the LiDAR–inertial odometry and path planning thread groups contain computationally intensive perception and planning workloads, whose WCETs are significantly larger and exhibit greater variability.

These task groups form the scheduling units controlled by the proposed scheduling policy, as described in Section 2.3. By operating at the task-group level, the scheduler adjusts execution frequencies without modifying internal algorithmic implementations or kernel-level scheduling mechanisms.

*Table 5.* Threads of the UAV system as scheduling targets.

| Thread | Description | Execution Time (ms) |
|---|---|---|
| **Flight Control Thread Group** | | |
| wq:vehicle_imu | IMU data reception and preprocessing | 0.010 |
| wq:vehicle_rate | Gyroscope data reception and preprocessing | 0.020 |
| wq:vehicle_acc | Accelerometer data reception and preprocessing | 0.004 |
| wq:vehicle_mag | Magnetometer data reception and preprocessing | 0.009 |
| wq:vehicle_air | Air pressure sensor data reception and preprocessing | 0.005 |
| wq:sensors | Aggregation and packaging of all sensor data | 0.012 |
| wq:ekf2 | Extended Kalman Filter state estimation | 0.062 |
| wq:mc_hover_est | Hover thrust estimator | 0.022 |
| wq:mc_rate_ctl | Angular rate closed-loop control | 0.013 |
| wq:mc_att_ctl | Attitude closed-loop control | 0.010 |
| wq:mc_pos_ctl | Position and velocity closed-loop control | 0.029 |
| wq:ctrl_alloc | Control allocation for motor thrust and torque | 0.026 |
| wq:pwm_out_sim | Motor command generation based on target motor speed | 0.018 |
| **LiDAR-Inertial Odometry Thread Group** | | |
| lio_lidar | LiDAR data reception and preprocessing | 0.057 |
| lio_imu | IMU data reception and preprocessing | 0.033 |
| lio_odom (lio_omp_0) | Mapping and odometry estimation | 3.335 |
| lio_omp_1 | OpenMP Thread | 3.335 |
| lio_omp_2 | OpenMP Thread | 3.335 |
| **Path Planning Thread Group** | | |
| ego_depthOdom | Depth image and odometry reception and synchronization | 0.203 |
| ego_updateOcc | Occupancy grid map update | 5.580 |
| ego_odometry | Odometry data reception and preprocessing | 0.006 |
| ego_execFSM | Finite state machine execution and mode switching | 1.112 |
| ego_checkColl | Trajectory collision checking and replanning | 2.788 |
| **Autonomous Flight Control Thread Group** | | |
| ego_traj | Trajectory waypoint generation from planned paths | 0.083 |
| ego_offboard | Conversion of waypoints to MAVLink position commands | 0.087 |

## C. Partially Observable Markov Decision Process

A Partially Observable Markov Decision Process (POMDP) provides a general framework for sequential decision making under uncertainty when the agent does not have direct access to the true underlying system state. Compared to a fully observable Markov Decision Process (MDP), a POMDP accounts for incomplete, noisy, or indirect observations of the environment, which arise naturally in many real-world systems (Kurniawati, 2022; Lauri et al., 2022).

We model the proposed UAV scheduling problem as an implicit POMDP. Formally, the problem is defined by the tuple

$$\langle \mathcal{S}, \mathcal{A}, \mathcal{T}, \mathcal{O}, \mathcal{Z}, \mathcal{R}, \gamma \rangle,$$

where $\mathcal{S}$ denotes the latent state space, $\mathcal{A}$ denotes the action space, $\mathcal{T}$ denotes the state transition process, $\mathcal{O}$ denotes the observation space, $\mathcal{Z}$ denotes the observation process, $\mathcal{R}$ denotes the reward function, and $\gamma$ denotes the discount factor.

**State Space $\mathcal{S}$.** In this work, the latent state $s_t \in \mathcal{S}$ represents the complete joint state of the physical UAV system and the onboard computing system at decision step $t$. It includes the vehicle pose, velocity, and surrounding environment state, sensor streams and buffers, internal states of the localization, planning, offboard-control, and flight-control modules, OS-level execution states such as task queues and timing behavior, resource states such as CPU usage, and the current scheduling configuration. These variables jointly determine future execution and flight behavior, but they are not fully available to the scheduler at runtime.

**Action Space $\mathcal{A}$.** The action $a_t \in \mathcal{A}$ specifies the execution frequencies assigned to the task groups controlled by the scheduling middleware,

$$a_t = (f_{\text{ctrl}}, f_{\text{loc}}, f_{\text{plan}}, f_{\text{ofb}}).$$

The flight-control frequency is selected from $\{100, 200, 300, 400\}$ Hz, and the localization, planning, and offboard-control frequencies are each selected from $\{10, 20, 30, 40, 50\}$ Hz. The resulting action space therefore contains $4 \times 5 \times 5 \times 5 = 500$ discrete scheduling configurations.

**Transition Process $\mathcal{T}$.** The transition process $\mathcal{T}(s_{t+1} \mid s_t, a_t)$ is induced by the coupled evolution of UAV dynamics, sensor generation, autopilot computation, and OS-level scheduling behavior under the selected frequency configuration. This process is stochastic because of sensing noise, environmental variation, execution-time variability, and OS-level timing jitter. We do not explicitly parameterize $\mathcal{T}$. Instead, the policy is learned directly from interaction data collected in the hardware-in-the-loop environment.

**Observation Space $\mathcal{O}$.** The observation $o_t \in \mathcal{O}$ is constructed by the runtime monitor and provides a lossy view of the latent state. Table 6 summarizes the monitored variables used by the policy.

*Table 6.* Observation variables used in the POMDP formulation.

| Category | Variable | Meaning |
|---|---|---|
| System | $u_{\text{ctrl}}$ | CPU time consumed by flight-control tasks |
| System | $u_{\text{nav}}$ | CPU time consumed by navigation tasks |
| Schedule | $f_{\text{ctrl}}$ | Current flight-control frequency |
| Schedule | $f_{\text{loc}}$ | Current localization frequency |
| Schedule | $f_{\text{plan}}$ | Current planning frequency |
| Schedule | $f_{\text{ofb}}$ | Current offboard-control frequency |
| Flight | $d_{\text{traj}}$ | Deviation from the planned trajectory |
| Flight | $v$ | Current vehicle velocity |
| Environment | $d_{\text{obs}}$ | Distance to the nearest obstacle |

**Observation Process $\mathcal{Z}$.** The observation process $\mathcal{Z}(o_t \mid s_t)$ maps the latent state to the runtime measurements available to the monitor. This mapping is partial because internal estimator states, planning states, task queues, and detailed timing behavior are not fully exposed to the policy. To mitigate this partial observability, the effective policy input is constructed from a fixed-length history of recent observations and scheduling actions with history length $H = 10$. The recurrent policy then learns a history-dependent scheduling strategy that approximates belief-state reasoning without requiring an explicit probabilistic model.

**Reward Function $\mathcal{R}$.** The reward $r_t \in \mathcal{R}$ is the weighted penalty function introduced in Section 3,

$$
\begin{aligned}
r_t = & - \lambda_{\mathrm{traj}} \, d_{\mathrm{traj}}\big(\mathcal{P}_t^{\mathrm{odom}}, \mathcal{P}_t^{\mathrm{ref}}\big) \\
& - \lambda_{\mathrm{est}} \, d_{\mathrm{traj}}\big(\mathcal{P}_t^{\mathrm{odom}}, \mathcal{P}_t^{\mathrm{gt}}\big) \\
& - \lambda_{\mathrm{cpu}} \, \mathbb{I}(u_t > 0.95) - \lambda_{\mathrm{col}} \, \mathbb{I}(\mathrm{collision}_t) \, .
\end{aligned}
$$

We set $\lambda_{\mathrm{traj}} = 0.01$, $\lambda_{\mathrm{est}} = 0.01$, $\lambda_{\mathrm{cpu}} = 0.1$, and $\lambda_{\mathrm{col}} = 1$.

**Discount Factor $\gamma$.** The discount factor is $\gamma = 0.99$, reflecting that scheduling decisions often affect flight performance through delayed estimation, planning, and control interactions.

At each time step, the scheduler selects actions based on observations or observation histories rather than direct access to the latent state (see Fig. 11). Since the true state is unobservable, classical POMDP methods often maintain an explicit belief state over $\mathcal{S}$. In our setting, the transition and observation models are not explicitly parameterized, so the recurrent policy uses observation-action histories to form an implicit latent representation for decision making.

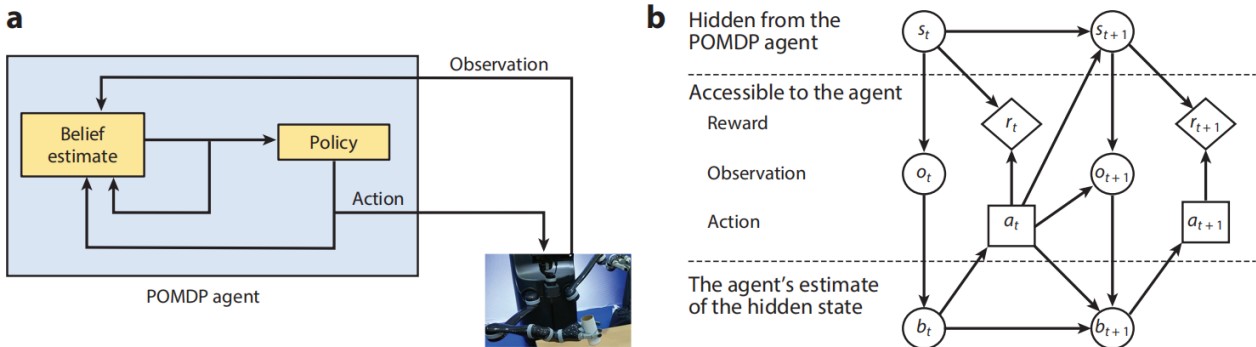

*Figure 11.* Illustration of a single time step of (a) a POMDP agent and (b) a POMDP process (Kurniawati, 2022).

# D. Reinforcement Learning for Control and Scheduling

Reinforcement learning (RL) provides a general framework for sequential decision making in dynamical environments, in which policies are optimized through interaction and long-term feedback. Its core objective is to optimize cumulative performance over time rather than instantaneous outcomes, making RL well suited to problems where actions influence future system behavior in a closed-loop manner.

This property is particularly relevant for control and scheduling, where the effects of decisions propagate through multi-stage computation and control pipelines and may only manifest after non-negligible delays. Policy-gradient and actor-critic methods directly optimize expected return and enable temporal credit assignment under such delayed effects (Sutton et al., 1998). Moreover, RL can operate under unknown or hard-to-model dynamics, which commonly arise in robotic systems due to unmodeled aerodynamics, friction, sensor noise, and software timing variability. By learning policies directly from interaction data, model-free RL can compensate for implicit disturbances and modeling errors (Lu et al., 2022). Finally, RL naturally accommodates partial observability through history-dependent policies. Recurrent architectures, such as LSTM-based actor-critic methods, maintain an internal memory over past observations, allowing the policy to reason about delayed and cumulative effects that cannot be inferred from instantaneous measurements alone. This capability is especially important for scheduling, where runtime metrics only partially reflect internal system states (Hausknecht & Stone, 2015).

RL has been widely applied to continuous-control problems in robotics, including locomotion, manipulation, and aerial vehicles. A recurring theme in these applications is robustness to nonlinearities and external disturbances. Policy-gradient methods have demonstrated the ability to stabilize nonlinear and chaotic systems without explicit analytical models, learning feedback behaviors that generalize across varying initial conditions and perturbations (Yau et al., 2024). In UAV domains, RL has been explored for guidance, trajectory tracking, and obstacle avoidance, especially in settings where environment complexity and safety objectives challenge purely model-based approaches (Kober et al., 2013; Lillicrap et al., 2020).

Scheduling problems can be viewed as sequential decision processes in which actions such as selecting execution rates or allocating computational resources influence future system load and downstream performance. These effects are often delayed and cross-coupled, making analytical modeling difficult and limiting the effectiveness of heuristic scheduling strategies. RL has therefore been investigated for system-level scheduling and resource management, including data-center scheduling and real-time task control, where policies adapt to workload variations and optimize long-term objectives such as latency and efficiency (Mao et al., 2016; Xu & Zhu, 2018).

In robotic systems, scheduling is tightly coupled with physical performance because perception, estimation, planning, and control form a computation pipeline whose timing properties directly affect closed-loop behavior. This coupling is particularly pronounced on embedded platforms with limited computational resources. In unified UAV architectures, scheduling decisions influence flight performance through delayed interactions among estimation accuracy, planning quality, and control tracking. Since these internal dynamics are only partially observable from runtime measurements, the resulting scheduling problem naturally benefits from history-dependent RL formulations.

Among modern RL algorithms, Proximal Policy Optimization (PPO) is commonly adopted in control and scheduling settings due to its stable on-policy updates and implementation simplicity (Schulman et al., 2017). PPO employs a clipped surrogate objective that constrains policy updates, reducing the risk of abrupt behavioral changes that could destabilize physical systems or induce oscillatory scheduling patterns. These properties are particularly desirable in embedded and hardware-in-the-loop settings, where software scheduling and physical dynamics interact in a nonstationary manner.

PPO supports continuous, discrete, and hybrid action spaces, and can be readily combined with recurrent architectures to address partial observability and delayed effects. As a result, recurrent PPO has been successfully applied to robotic control tasks, including UAV guidance and tracking, where temporal context and stable learning dynamics are critical (Duoxiu et al., 2022). These characteristics make PPO a pragmatic choice for learning history-dependent scheduling policies that regulate computational resources while preserving real-time feasibility and flight stability.

# E. Training Hyperparameters

Table 7 summarizes the main optimization hyperparameters used to train the scheduling policy. Table 8 reports the policy architecture and input-output configuration.

*Table 7.* Training hyperparameters used for Recurrent PPO.

| Item | Value |
|------|-------|
| Algorithm | RecurrentPPO |
| Policy | MlpLstmPolicy |
| Learning rate | $3.0 \times 10^{-4}$ |
| `n_steps` | 10 |
| Batch size | 10 |
| `n_epochs` | 10 |
| Discount factor $\gamma$ | 0.99 |
| GAE $\lambda$ | 0.95 |
| Clip range | 0.2 |
| Clip range for value function | None |
| Entropy coefficient | 0 |
| Value loss coefficient | 0.5 |
| Maximum gradient norm | 0.5 |
| Advantage normalization | True |
| Target KL | None |
| State-dependent exploration | False |
| SDE sample frequency | $-1$ |
| Optimizer | Adam |
| Adam betas | $(0.9, 0.999)$ |
| Adam epsilon | $1.0 \times 10^{-5}$ |
| Number of environments | 1 |
| Environment step rate | 10 Hz |

*Table 8.* Policy architecture and input-output configuration.

| Item | Value |
|------|-------|
| Observation space | $\text{Box}(80, )$, `float32` |
| Action space | $\text{MultiDiscrete}([4, 5, 5, 5])$ |
| Feature extractor | FlattenExtractor |
| Recurrent core | Separate actor and critic LSTMs |
| LSTM layers | 1 |
| LSTM hidden size | 256 |
| Actor MLP | $64 \rightarrow 64$ with Tanh activations |
| Critic MLP | $64 \rightarrow 64$ with Tanh activations |
| Action head | $\text{Linear}(64 \rightarrow 19)$ logits split into 4 categorical branches |

## F. Hardware-in-the-Loop Simulation

The simulator is responsible for physical dynamics, virtual sensing, and actuation, while the UAV system executes the complete autopilot software stack, processes incoming sensor streams, and computes control commands.

The simulator runs on a high-performance PC or workstation and continuously exchanges data with the UAV system. Virtual sensor measurements, such as point clouds and depth images, are streamed to the onboard computer, while actuator commands produced by the autopilot are sent back to the simulator to drive the simulated vehicle. This separation allows computationally intensive physics simulation to be performed offboard, while preserving realistic onboard execution, timing behavior, and system-level interactions.

The simulator is implemented using Gazebo Classic (Koenig & Howard, 2004) and consists of a physics engine, simulator plugins, and virtual 3D models. We retain the default Open Dynamics Engine (ODE) (Smith, 2006) as the physics backend. In this work, our development primarily focuses on simulator plugins and system models, rather than modifying the underlying physics engine, enabling flexible integration with different autopilot configurations.

To support HIL execution, multiple plugins are deployed to establish a closed simulation loop. Virtual sensor plugins, including LiDAR (Vultaggio et al., 2023) and depth camera (pal-robotics, 2018), generate synthetic measurements that initiate the dataflow. These measurements are transmitted to the UAV system through ROS and MAVLink, where the autopilot tasks perform state estimation, planning, and control. The resulting actuator commands are then sent back to the simulator and applied by motor plugins, which generate thrust and torque inputs that are processed by the physics engine.

The virtual 3D models include the UAV model, environmental objects, and the simulation "world" constructed from them. Our framework initially provides six representative world environments (see Fig. 12), such as village, urban, and forest scenes, to support diverse evaluation scenarios. In addition to the provided environments, users can extend the simulator by adding custom object models or constructing new worlds following the standard Gazebo modeling workflow.

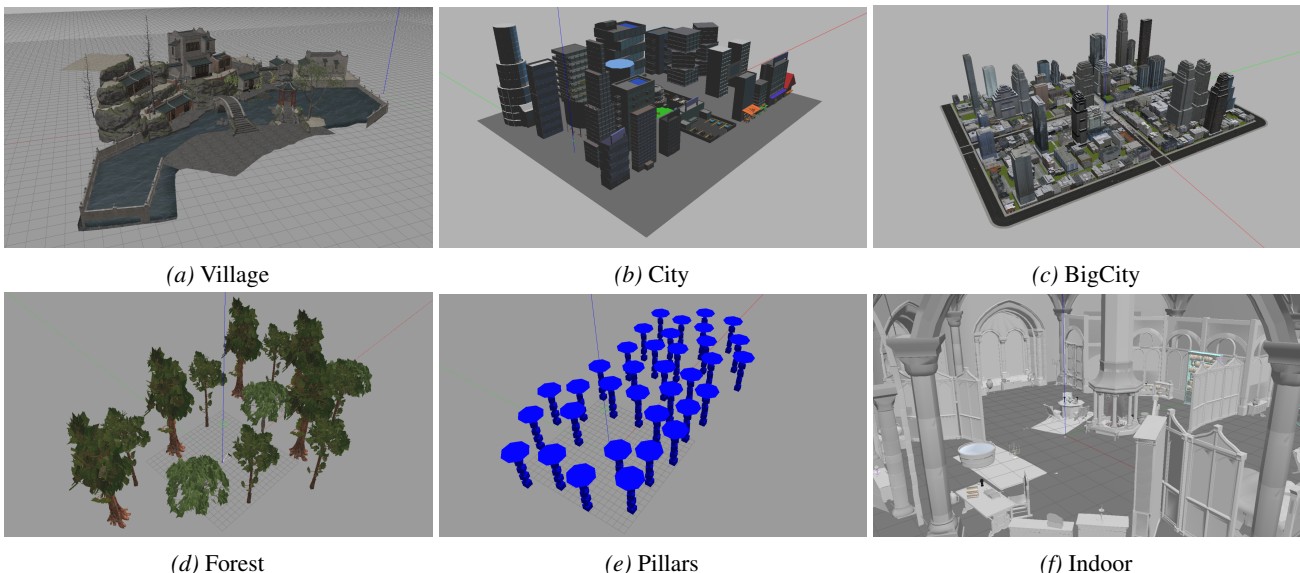

*(a)* Village      *(b)* City      *(c)* BigCity

*(d)* Forest      *(e)* Pillars      *(f)* Indoor

*Figure 12.* Simulation "worlds" initially provided by the framework.

We use ROS1 and Gazebo Classic because the HIL workflow in this study relies on existing UAV autopilot tools, simulator plugins, and MAVLink-based interfaces that are mature in this software stack. This choice is an implementation decision rather than a methodological requirement. The proposed scheduling formulation and middleware design can be migrated to ROS2 and Ignition Gazebo-based systems, provided that equivalent MAVLink interfaces, HIL simulator plugins, runtime monitoring signals, and schedulable task abstractions are exposed in the target software stack.

## G. Cross-Scenario Generalization

To further examine whether the learned scheduling policy is tied to the training scenario, we evaluate it in additional simulation environments without transfer, fine-tuning, or retraining. The policy is tested in City, Village, and Sparse-Pillars environments using the same baselines and metrics as in Section 5. These environments differ in obstacle density, scene layout, and navigation geometry, so they provide a broader test of whether the learned frequency-adaptation behavior remains effective beyond the original evaluation scene.

Figs. 13, 14, and 15 show the resulting trajectory deviation, CPU time, and success rate. Across the three environments, the

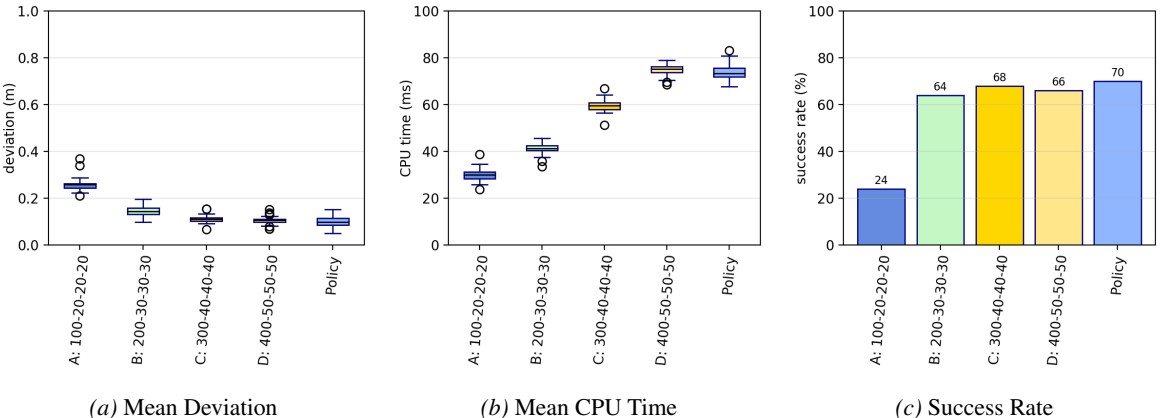

*(a)* Mean Deviation  *(b)* Mean CPU Time  *(c)* Success Rate

*Figure 13.* Cross-scenario evaluation results in the City environment.

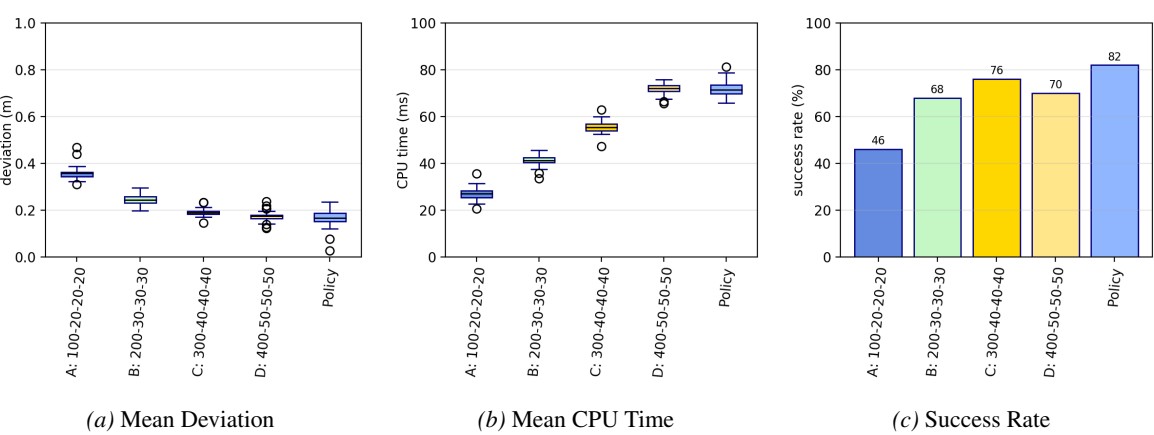

*(a)* Mean Deviation  *(b)* Mean CPU Time  *(c)* Success Rate

*Figure 14.* Cross-scenario evaluation results in the Village environment.

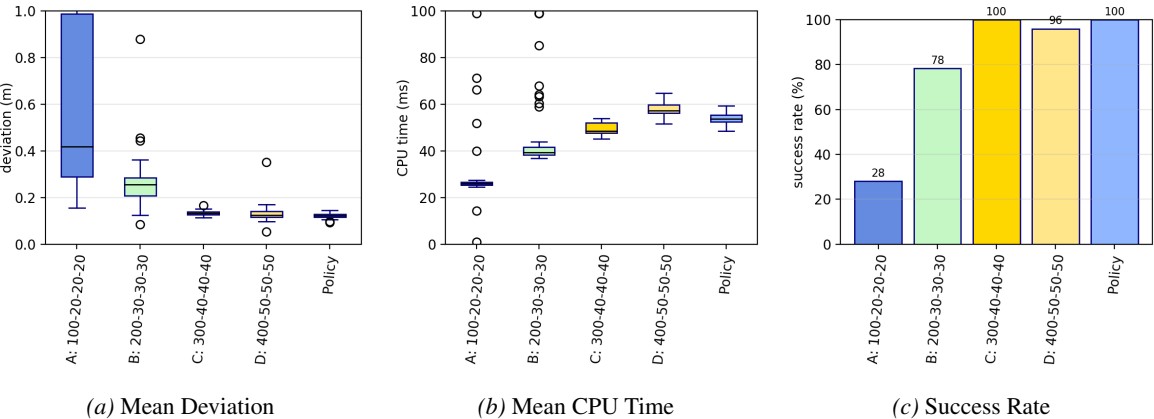

*(a)* Mean Deviation  *(b)* Mean CPU Time  *(c)* Success Rate

*Figure 15.* Cross-scenario evaluation results in the Sparse-Pillars environment.

learned policy generally maintains competitive trajectory tracking while achieving strong mission robustness relative to fixed-rate baselines. The advantage is most visible in success rate, indicating that adaptive scheduling helps reduce failure cases that are not fully captured by mean deviation alone. In the easier Sparse-Pillars environment, the performance gap becomes smaller because the reduced obstacle density leaves less room for scheduling decisions to affect end-to-end flight outcomes. Overall, these results suggest that the learned scheduler captures reusable runtime adaptation patterns.

## H. Policy Behavior and Failure Cases

We further inspect representative in-flight scheduling traces to better understand the behavior learned by the policy. As shown in Fig. 16, the policy does not simply select the highest available frequencies. During faster flight in relatively open space, it tends to allocate higher frequencies to localization and offboard control, which improves state feedback and command updates while avoiding unnecessary planning load. When the vehicle approaches obstacles, the policy tends to increase the planning frequency, improving the responsiveness of collision checking and replanning.

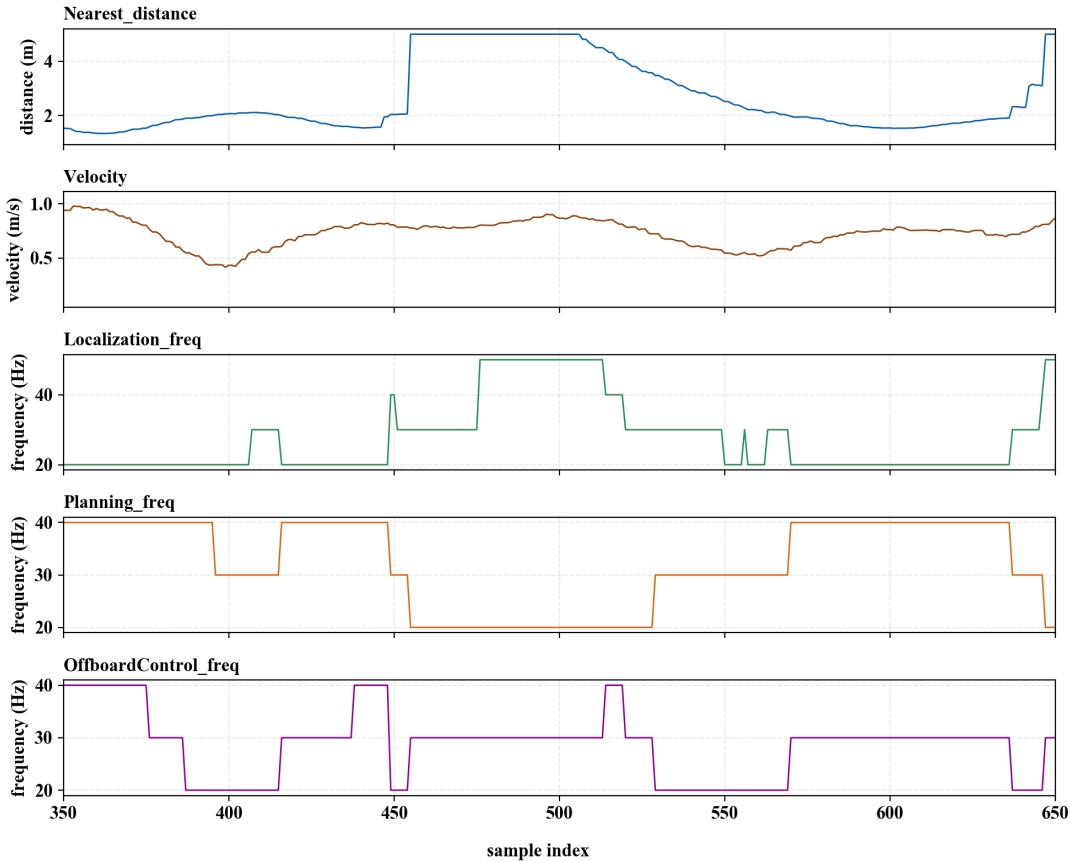

*Figure 16.* Representative learned scheduling behavior during flight.

The remaining failures are mainly associated with delayed recovery of planning frequency under rapidly changing obstacle proximity. In the representative collision episode shown in Fig. 17, the policy first reduces the planning frequency to allocate more computation to localization, but does not raise it quickly enough when another obstacle becomes close. This delay postpones avoidance updates and eventually leads to collision. This behavior highlights a limitation of the current frequency-level scheduler. The bounded action space and terminal collision penalty are engineering safeguards, but they do not constitute a formal safety guarantee. Deployment in safety-critical settings should therefore combine the learned scheduler with conservative runtime monitors, fallback scheduling rules, and platform-specific safety constraints.

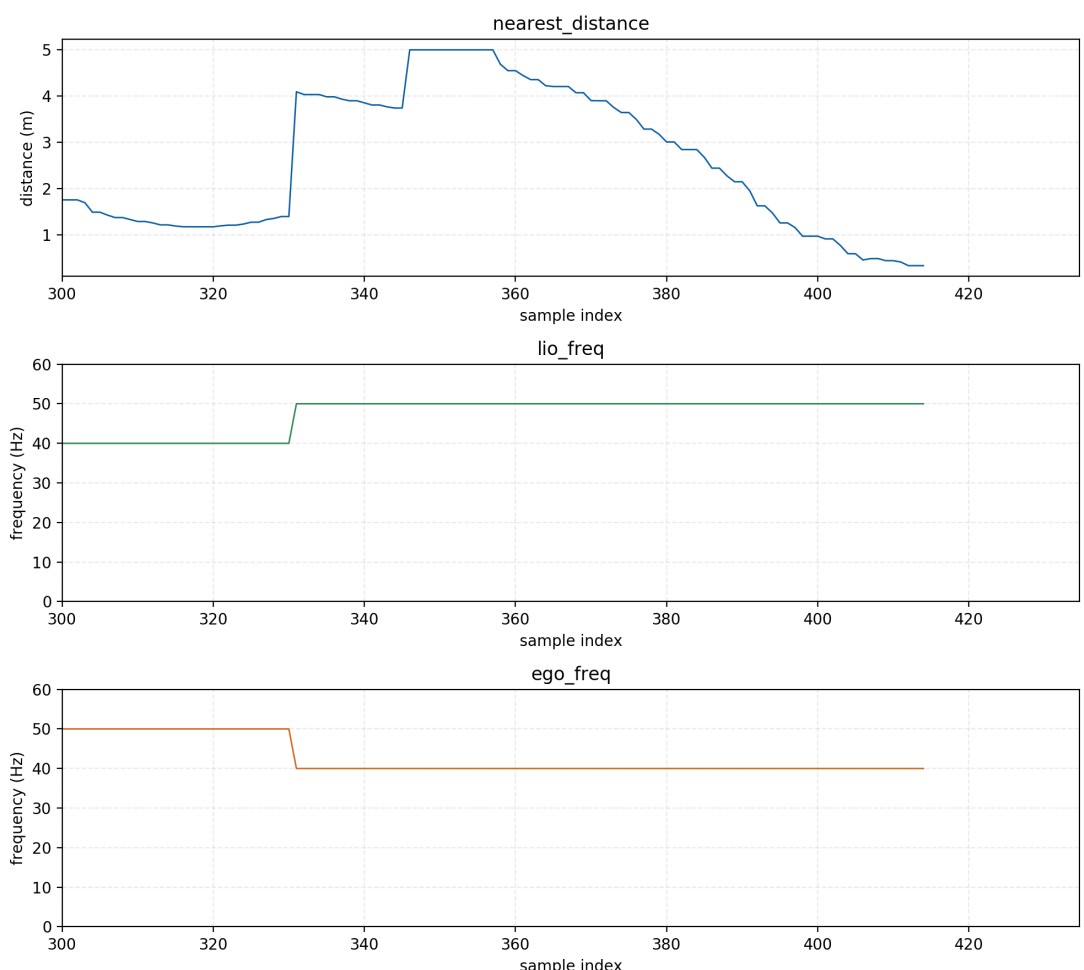

*Figure 17.* Representative failure case caused by delayed planning-frequency recovery near obstacles.

# I. Cross-Hardware Transfer

Scheduling policies are inherently hardware-dependent because the effect of a frequency decision depends on processor speed, core allocation, operating-system timing, and task execution costs. To examine whether the proposed framework can support adaptation to a different compute budget, we conduct a preliminary cross-hardware transfer experiment. The original policy is evaluated under the target hardware setting and then fine-tuned using the same scheduling interface and reward structure. The transferred policy is compared with the fixed-rate baselines and the original policy.

Fig. 18 reports the resulting trajectory deviation, CPU time, and success rate. Directly applying the original policy to the target hardware setting yields degraded robustness, confirming that zero-shot transfer across hardware configurations is limited. After transfer training, the policy improves the success rate from $46\%$ to $62\%$ and reduces the trajectory deviation relative to the original policy, while maintaining a comparable CPU-time profile. These results suggest that the proposed framework can be adapted to a new hardware configuration through additional training, although deployment on substantially different SoCs should not assume zero-shot generalization.

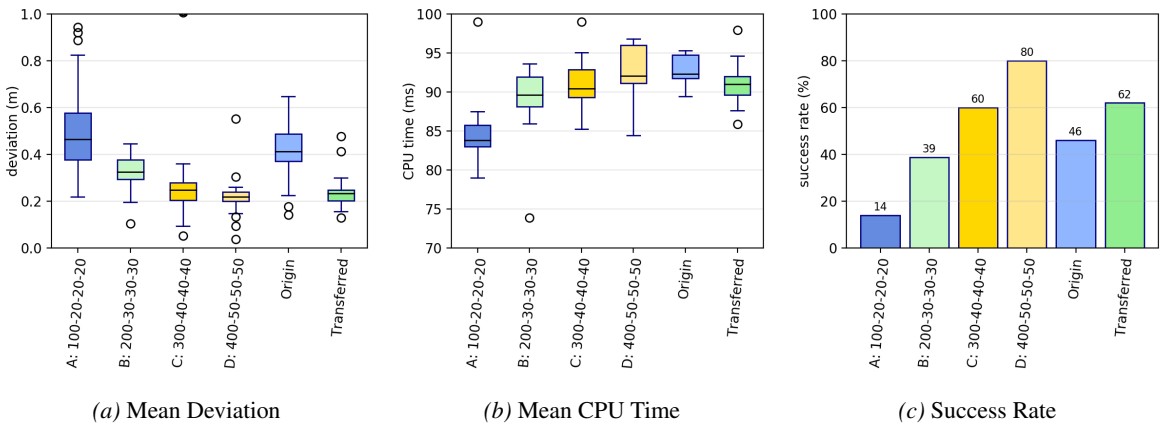

*(a)* Mean Deviation  *(b)* Mean CPU Time  *(c)* Success Rate

*Figure 18.* Preliminary cross-hardware transfer results.

## J. Unified UAV platform

The unified UAV platform studied in this work has been developed and validated through long-term software and hardware iteration, as well as extensive real-world flight experiments and deployments.

As illustrated in Fig. 19, the UAV payload and onboard processing unit consist of a single card-sized onboard computer, which executes all navigation, perception, planning, and flight-control tasks. Unlike conventional split architectures, no dedicated flight control unit is used. To support reliable flight control on the onboard computer, we design a custom sensor expansion board that provides essential sensing capabilities, including an IMU, magnetometer, and barometer. The sensor board communicates with the onboard computer via standard SPI and I$^2$C buses, and adopts rigid board-to-board connectors to improve signal integrity and communication robustness. The structure and functionality of the sensor board are also shown in Fig. 19.

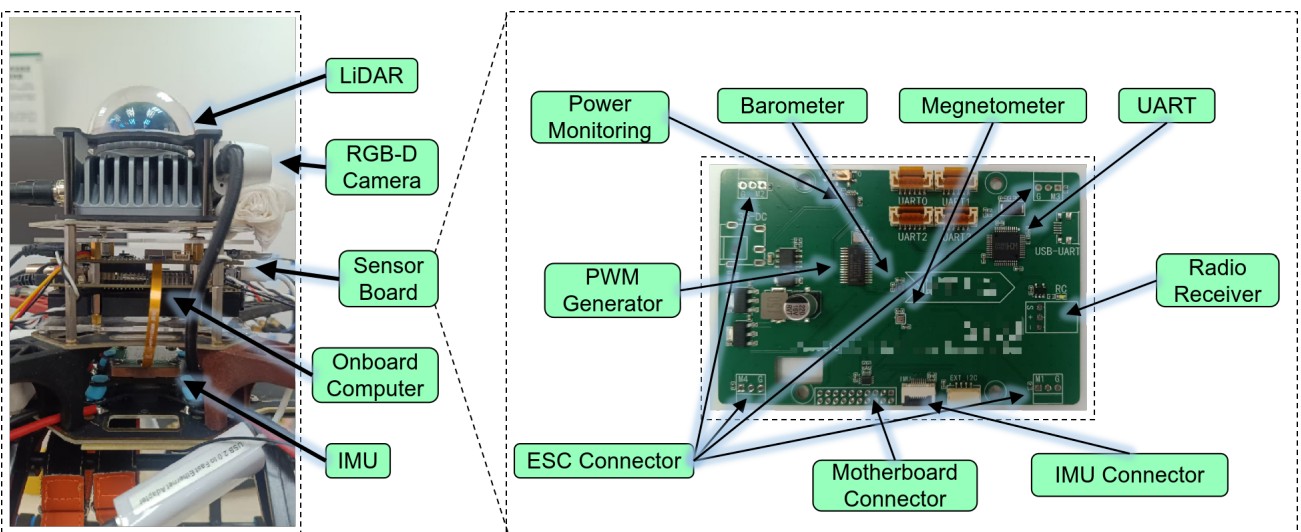

*Figure 19.* Hardware design of a unified UAV system.

Figure 20 presents a hardware component selection example of the UAV system. The platform enables fully onboard autonomous capabilities, including local mapping, state estimation, obstacle avoidance, trajectory planning, and autonomous flight.

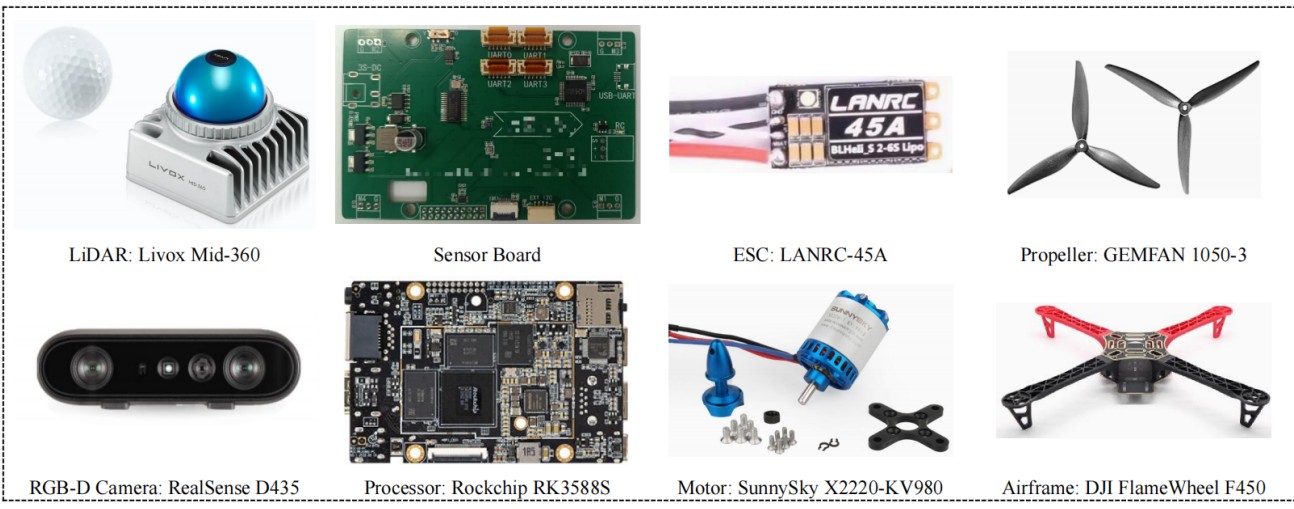

*Figure 20.* Hardware components of a unified UAV system prototype.

The fully assembled UAV has a total weight of 2.2 kg and supports a maximum payload of 2.3 kg (see Fig. 21), with a typical flight endurance of approximately 20 minutes.

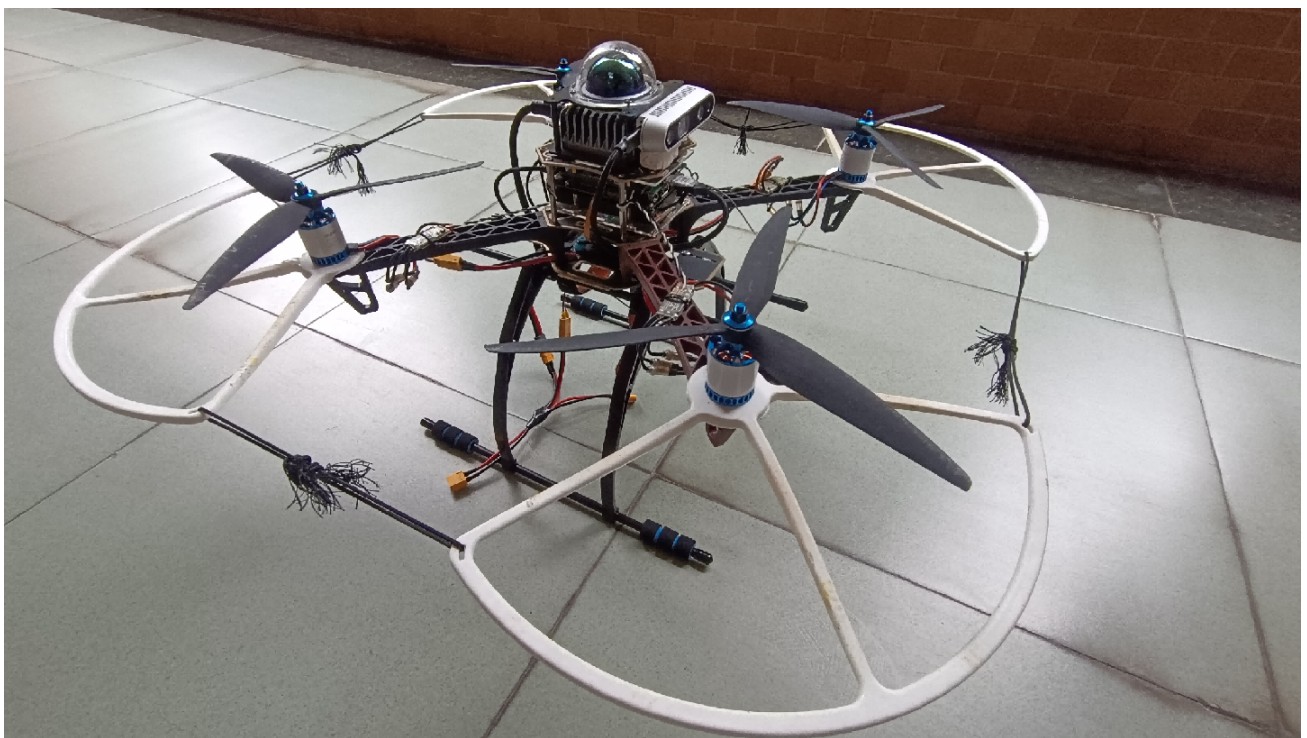

*Figure 21.* Overview of UAV.

