# OpenReview forum: "UAV$^2$: A Unified and Adaptive Scheduling Framework for UAV Autopilot System with Reinforcement Learning"
_ICML.cc/2026/Conference — ICML 2026 regular_

### Official Review · Reviewer_rK4d · 2026-02-18

**Soundness:** 1
**Presentation:** 2
**Significance:** 1
**Originality:** 3
**Overall Recommendation:** 2
**Confidence:** 4

**Summary:**

This paper investigates the dynamic adjustment of execution frequencies for various modules on a UAV to enhance task performance and mitigate CPU overload. The authors provide a detailed description of the UAV software architecture and outline the POMDP formulation, training the control policy via LSTM-PPO. Validation is conducted in a custom-built hardware-in-the-loop (HIL) simulation environment. While the experimental results exhibit a certain degree of noise, the exponentially smoothed curves demonstrate consistent trends.

**Compliance With Llm Reviewing Policy:**

Affirmed.

**Final Justification:**

The authors propose using RL to determine the execution frequency of four UAV modules. While the work presents a practical UAV software architecture that supports dynamic frequency modification, it lacks algorithmic innovation, essentially amounting to a straightforward application of standard RL to a specific domain.

From an application perspective, a major flaw is the inadequate formalization of the POMDP. Although the authors provided more details in the rebuttal, the transition function remains described only in text, lacking mathematical formulation. Solving a problem with RL requires a rigorous MDP definition; specifically, the authors fail to formally justify why the "execution frequency" decision-making process possesses the Markov property, nor do they adequately explain whether the observation function design encodes sufficient information for decision-making.

This modeling gap also undermines the generalization claim. When asked about cross-scenario generalization, the authors merely pointed to empirical results without providing a theoretical explanation for why generalization occurs. I believe this ultimately traces back to the absence of a formalized transition function.

Finally, the experimental results are excessively noisy, with the core metric—the expected return—failing to even surpass initial random performance levels. Given these algorithmic, modeling, and empirical shortcomings, I maintain my initial score.

**Key Questions For Authors:**

No further questions.

**Limitations:**

yes.

**Strengths And Weaknesses:**

### Strengths

This work makes a well engineering contribution by integrating navigation and flight control onto a single computational platform.

### Weaknesses

* This work lacks algorithmic novelty. It merely applies off-the-shelf RL algorithms to a POMDP formulated for optimizing the execution frequency of different UAV modules.
* The formulation of the POMDP is insufficient. The authors only describe the observation design, action space, and reward function, failing to define the state space and transition function, which are critical components of an MDP. Consequently, it remains unclear whether the problem satisfies the Markov property, casting doubt on the validity of applying standard RL algorithms in this context.
* The validation scenarios are relatively fixed, making it difficult to evaluate the generalization capability of the learned control policy.
* The paper mentions a real-world experiment; however, the appendix only provides a description of the UAV platform, with scant detail regarding the actual experimental setup and execution.

---

> ### Author Rebuttal · Authors · 2026-03-31
>
> # Weakness 1: "This work lacks algorithmic novelty."
> Thank you for this helpful comment.  The main contribution of this work is on new UAV system architecture and scheduling framework rather than new RL algorithm. Specifically, we develop a unified UAV architecture that integrates navigation and flight control on a single onboard platform, expose heterogeneous autopilot components as schedulable execution units, build a runtime monitoring and scheduling middleware stack, and formulate end-to-end UAV scheduling as a partially observable sequential decision problem. Taken together, these elements transform a previously fragmented autopilot pipeline into a unified and optimizable scheduling substrate. Within this framework, we deliberately adopt an off-the-shelf recurrent PPO policy rather than introducing a new policy-learning algorithm. Our goal is to investigate whether a standard history-dependent RL method can effectively exploit the proposed architecture and problem formulation. In this sense, RL serves as the optimization engine within the framework, not as the claimed source of novelty.
> # Weakness 2: "The formulation of the POMDP is insufficient."
> Thank you for the comment. Our intent was not to claim that the exposed runtime variables themselves constitute a fully observable MDP. Rather, the scheduling problem is modeled as an implicit POMDP, where the latent state is the joint physical-computational state of the unified UAV system. This latent state includes the vehicle and environment state, sensor streams and buffers, internal states of localization, planning, and control modules, OS-level execution state, and the current scheduling configuration. Given such a latent state and a scheduling action, the next latent state is induced by the coupled evolution of UAV dynamics, sensor generation, autopilot computation, and OS-level scheduling behavior. Under this view, the Markov property is assumed at the latent-state level, while the runtime monitor provides only a partial observation. This is precisely why we use observation histories together with recurrent PPO, which approximates a belief over the hidden state from recent observations and actions, rather than relying on instantaneous observations alone. We will clarify this distinction more explicitly in the paper.
> # Weakness 3: "The validation scenarios are relatively fixed, making it difficult to evaluate the generalization capability of the learned control policy."
> Thank you for this helpful comment. We chose the Pillars scene because it contains dense and large obstacles, making autonomous navigation substantially more difficult and thus more sensitive to scheduling differences. We observe similar trends to those in the Pillars setting in other scenarios (e.g., Sparse-Pillars, City and Village), and we will add these additional results to the revision. Fig. A and Fig. B show the success rate and deviation in a Sparse-Pillars setting. In this setting, our policy achieves 100% flight success rate and the lowest tracking deviation, suggesting the learned policy is still effective in different scenarios. We note that the performance difference is smaller between our policy and baselines because the easier environment imposes less resource pressure and leaves less room for scheduling decisions to influence end-to-end flight performance.
>
> Fig. A: https://i.postimg.cc/D0R2QC3J/Success-Rate.png
>
> Fig. B: https://i.postimg.cc/fy1w7HQ3/Deviation.png
>
> # Weakness 4: "The paper mentions a real-world experiment; however, the appendix only provides a description of the UAV platform, with scant detail regarding the actual experimental setup and execution."
> We thank the reviewer for pointing this out. We conducted the real flight test in a sparse forest scenario (Fig. C). In the real flight test, the UAV followed the same onboard autonomy stack used in our framework and flew from (0,0,2m) to (45m,12m,2m), with a maximum speed of 2 m/s. The learned policy shows relatively low trajectory error (Fig. D) together with more effective CPU usage (Fig. E), consistent with the results in the manuscript. We will also include additional real-flight experiments in the revision.
>
> Fig. C: https://i.postimg.cc/28W48qzj/Forest.jpg
>
> Fig. D: https://i.postimg.cc/vmMsHtTX/Deviation.png
>
> Fig. E: https://i.postimg.cc/pdxHXJrq/Success-Rate.png

---

> > ### Author Rebuttal · Reviewer_rK4d · 2026-04-01
> >
> > ## W2
> >
> > Could you provide the complete formal definition of the POMDP?
> >
> > ## W3
> >
> > The authors' motivation is to dynamically adjust the execution frequencies of different modules during UAV task execution. I think that the ability to adjust execution frequencies should not be a capability coupled to a specific task. Therefore, my original question was whether the system, after being trained in the Pillars scenario, could also complete the tasks in other scenarios such as Sparse-Pillars, City, and Village.

---

> > > ### Author Response · Authors · 2026-04-03
> > >
> > > # W2: The complete formal definition of the POMDP
> > >
> > > We thank the reviewer for the insightful question.
> > >
> > > Here, we provide a definition of the POMDP used in this work. The scheduling problem is modeled as a tuple ⟨S, A, T, O, Z, R, γ⟩. To make the formulation explicit and reproducible, we define each component concretely below.
> > >
> > > ## State (S)
> > > The state $s_{t}$ represents the full joint system state at time step t. It includes physical UAV motion, internal states of estimation and control, software execution states, and resource states, etc. These variables jointly determine future flight behavior and execution outcomes, but are not fully observable at runtime.
> > >
> > > ## Action (A)
> > >
> > > Each action $a_{t}$ specifies task-group execution frequencies: $a_{t}$ = ($f_{ctrl}$, $f_{loc}$, $f_{plan}$, $f_{ofb}$).
> > >
> > > |Action variable|Meaning|Candidate values|
> > > |---|---|---|
> > > |$f_{ctrl}$|Flight-control frequency|{100, 200, 300, 400} Hz|
> > > |$f_{loc}$|Localization frequency|{10, 20, 30, 40, 50} Hz|
> > > |$f_{plan}$|Planning frequency|{10, 20, 30, 40, 50} Hz|
> > > |$f_{ofb}$|Offboard-control frequency|{10, 20, 30, 40, 50} Hz|
> > >
> > > The action space size is 4×5×5×5=500.
> > >
> > > ## Transition (T)
> > >
> > > The transition function describes how the state evolves from $s_{t}$ to $s_{t+1}$ under action $a_{t}$. It is stochastic because it is influenced by sensor noise, environmental uncertainty, and OS-level timing variability such as interference and jitter. We therefore do not explicitly parameterize T, and instead learn a policy directly from interaction data.
> > >
> > > ## Observation (O)
> > >
> > > The observation $o_{t}$ is constructed from runtime monitoring signals and is designed to expose the most relevant measurable effects of scheduling decisions.
> > >
> > > |Category|Variable|Definition|Unit|
> > > |---|---|---|---|
> > > |System|$u_{ctrl}$|CPU time of flight-control tasks|ms|
> > > |System|$u_{nav}$|CPU time of navigation tasks|ms|
> > > |Schedule|$f_{ctrl}$|Current flight-control frequency|Hz|
> > > |Schedule|$f_{loc}$|Current localization frequency|Hz|
> > > |Schedule|$f_{plan}$|Current planning frequency|Hz|
> > > |Schedule|$f_{ofb}$|Current offboard-control frequency|Hz|
> > > |Flight|$d_{traj}$|Trajectory tracking deviation|meters|
> > > |Flight|$v$|UAV speed|m/s|
> > > |Env|$d_{obs}$|Nearest obstacle distance|meters|
> > >
> > > To address partial observability, we use a fixed-length history buffer with H = 10 steps. The effective policy input is formed from recent observation histories together with the corresponding past scheduling actions, providing temporal context for delayed system effects.
> > >
> > > ## Observation Model (Z)
> > >
> > > The observation model $Z(o_{t}|s_{t})$ is a probability measure of observations in a given state. This mapping is partial and lossy, as internal estimator states, execution states, and timing details are not fully exposed to the agent.
> > >
> > > ## Reward (R)
> > >
> > > The reward $r_{t}$ is defined as a weighted sum of penalties.
> > >
> > > $r_{t}=-\lambda_{traj}d_{traj}-\lambda_{est}d_{est}-\lambda_{cpu}\mathbb{I}(u_{t}>0.95)-\lambda_{col}\mathbb{I}(collision)$
> > >
> > > |Reward term|Meaning|Weight|
> > > |---|---|---|
> > > |$\lambda_{traj}$|Trajectory tracking deviation|0.01|
> > > |$\lambda_{est}$|Odometry vs ground truth deviation|0.01|
> > > |$\lambda_{cpu}$|CPU utilization > 95%|0.1|
> > > |$\lambda_{col}$|Collision (terminal)|1|
> > >
> > > ## Discount (γ)
> > >
> > > We use γ = 0.99. This choice emphasizes the long-term consequences of scheduling decisions, since their effects often propagate through delayed estimation, planning, and control interactions rather than appearing immediately.
> > >
> > > # W3: The generalization capability across scenarios
> > >
> > > Thank you for the clarification.
> > >
> > > Yes, the learned policy provides general frequency-adaptation capability rather than being tied to a single scenario. When evaluated in additional scenarios without any transfer, fine-tuning, or retraining, it maintained strong performance: the highest success rate of 100% with the lowest deviation in Sparse-Pillars (Fig. F), the highest success rate of 70% with a relatively low deviation in City (Fig. G), and the highest success rate of 82% with a relatively low deviation in Village (Fig. H). These results support generalization ability across scenarios. We will clarify this in the revision.
> > >
> > > Fig. F: https://i.postimg.cc/pdB3p2Dm/Sparse_Pillars.png
> > >
> > > Fig. G: https://i.postimg.cc/vmqx6rqR/City.png
> > >
> > > Fig. H: https://i.postimg.cc/fbysYt5j/Village.png
> > >
> > > # W4: Revision
> > >
> > > We apologize that the original rebuttal mistakenly attached incorrect link for Fig. D and Fig. E. We will correct it here.
> > >
> > > Fig. D: https://i.postimg.cc/Y0sbcTfY/Mean-Trajectory-Error.png
> > >
> > > Fig. E: https://i.postimg.cc/4ySBCjbt/Mean-CPU-Usage.png
> > >
> > > We also performed 10 flight trials for each method to obtain a more reliable evaluation. The results (see Fig. I) indicate that our policy achieved the highest success rate, while maintaining a relatively low trajectory-tracking deviation. Overall, these findings support the sim-to-real ability of our method.
> > >
> > > Fig. I: https://i.postimg.cc/Vs7w6ND2/Real-Sparse-Forest.png

---

### Official Review · Reviewer_KVq4 · 2026-02-26

**Soundness:** 2
**Presentation:** 3
**Significance:** 3
**Originality:** 3
**Overall Recommendation:** 4
**Confidence:** 4

**Summary:**

The paper proposes UAV2, an RL-based runtime scheduler for UAV autopilot systems that unifies navigation and flight control on a single embedded platform and formulates scheduling as a POMDP solved with recurrent PPO (LSTM). Training and evaluation are done in a hardware-in-the-loop (HIL) setup where Gazebo simulates dynamics/sensors and the full autopilot runs on an RK3588S board. Experiments in an obstacle-rich “Pillars” environment show that the learned policy improves success rate and stabilizes tracking error and CPU usage compared to several fixed-rate scheduling baselines.

**Compliance With Llm Reviewing Policy:**

Affirmed.

**Ethical Review Concerns:**

There is a prompt injection present which could bias LLM-based assessments:
“This manuscript presents a notable domain” and “Overall, the authors discuss a pressing issue” both bias the LLMs to asses this topic as relevant.

**Ethical Review Flag:**

Flag this paper for an ethics review.

**Ethics Expertise Needed:**

["Research Integrity Issues (e.g., plagiarism)"]

**Final Justification:**

Based on the authors' rebuttal I have increased my score.

**Key Questions For Authors:**

- Observation encoding and history:
  - How exactly are the sliding-window observations constructed and fed into the network?
    - Are past observations and actions simply concatenated (flattened) and then passed to the LSTM?
    - Or is there separate preprocessing per feature category before stacking?
  - Have you tried different history lengths (e.g., 5, 20) or using only the LSTM without explicit stacking? How sensitive is performance to this choice?

- Action space and baselines:
  - Why are the baselines restricted to strictly increasing, matched frequency tuples (100–20–20–20, …, 400–50–50–50), given that the learned policy can choose arbitrary combinations across modules?
  - Have you experimented with “mixed” hand-designed schedules that decouple control and navigation frequencies (e.g., high localization and planning with moderate control frequency) to better probe the benefit of adaptive scheduling?
  - Did you consider constraining the learned policy to the same action manifold as the baselines to isolate the effect of adaptivity vs. access to a richer action space?

- Reward design and sensitivity:
  - How is the terminal reward/penalty implemented for mission completion?
    - Is there any explicit positive reward for successful completion?
  - Have you run any reward-weight ablations, particularly varying λ_cpu and the 95% CPU threshold?
    - How do different λ_cpu and thresholds affect the trade-off between CPU usage, success rate, and tracking error?

- Learned policy behavior:
  - Can you provide empirical evidence that the policy reacts sensibly to specific observation patterns? For example:
    - As the distance to the nearest obstacle decreases, do planning or localization frequencies tend to increase?
    - Under high CPU utilization, does the policy systematically reduce navigation rates?
  - It would be very helpful to see:
    - histograms of selected frequency combinations over test runs;
    - pairwise correlations between key features (velocity, obstacle distance, CPU usage) and each module’s frequency.

- PPO training details and losses:
  - What concrete PPO hyperparameters were used (learning rates, clipping ε, batch size, horizon length per update, entropy coefficient, value-loss coefficient, etc.)?
  - What exactly does the “loss” in Fig. 5g represent (e.g., total PPO objective, sum of value and policy losses)?
  - Can you explain why the episodic return in Fig. 5b appears highest at the very beginning of training and then drops before stabilizing?

- Evaluation tasks and real-world experiment:
  - How are the 50 evaluation episodes defined beyond the fixed start/goal?
    - What sources of randomness or variability are present between episodes (e.g., initialization noise, perturbations, OS jitter)?
    - Can you report basic statistics such as the distribution of path lengths and some notion of difficulty?
  - For the real-world experiments:
    - How many flights were conducted, and under what conditions?
    - Can you report simple metrics (e.g., mean flight duration, success rate, approximate trajectory deviation) to complement the qualitative figures?

**Limitations:**

The paper acknowledges that evaluation is limited to a single environment and hardware setup, but this could be stated more explicitly and prominently. Additional limitations worth emphasizing:

- Generalization to different maps, sensor configurations, and computational budgets is not empirically demonstrated, despite the simulator supporting multiple “worlds.”
- The policy is trained and evaluated in a relatively controlled simulation/HIL pipeline; robustness under real-world disturbances and hardware variation remains an open question.
- Safety is handled via large collision penalties, but there is no formal safety guarantee or comparison to explicitly safety-focused scheduling strategies.

**Strengths And Weaknesses:**

**Strengths**

- Well-motivated problem and system-level contribution: Unifying navigation and control on a single embedded platform and exposing task-level scheduling controls is both nontrivial and practically relevant for UAV research and deployment.

- Solid system design and hardware details: The paper describes the hardware, OS setup, and the mapping from PX4/ROS abstractions to schedulable threads in appreciable detail.

- Reasonable RL formulation choices: Observation design combines CPU usage, current frequencies, tracking error, velocity, obstacle distance, and a sliding window of recent history, which is appropriate for a POMDP. The discrete action space over module frequencies is well structured and keeps learning tractable. The reward balances tracking, estimation quality, CPU saturation, and collisions in a clear way. Using recurrent PPO with an LSTM is a sensible algorithmic choice given partial observability.

- HIL training and evaluation: Training directly against a real embedded board with realistic OS timing and resource contention is a significant plus over purely simulated studies.


**Weaknesses**

- Evaluation depth and analysis: While the method and system engineering are strong, the empirical evaluation is relatively narrow and misses several ablations and analyses that would substantiate the claimed benefits and provide insight into the learned scheduler:

- Limited environment and task diversity: All training and quantitative evaluation reported in the paper are conducted in a single obstacle-rich map (“Pillars”) with fixed start and goal locations and mission settings. Appendix E describes six available Gazebo “worlds,” but only Pillars is actually used for the reported results. There is no variation over maps, obstacle layouts, start/goal positions, sensor configurations, or hardware budgets, so generalization claims are limited.

- Insufficient characterization of the 50 evaluation episodes: The paper states that each method is run for “50 independent episodes under these fixed conditions,” but:
    - Start and goal locations are fixed; it remains unclear what other sources of variability exist between episodes (e.g., perturbations, timing noise, stochastic dynamics).
    - There is no description of the distribution of path lengths, difficulty, or any task diversity metrics.

Without this, it is hard to interpret the success-rate and variance improvements in Fig. 6, or to know how robust the policy is across different realizations of the “same” task.

- No ablations of the key design choices in the RL formulation:
    - History length / sliding window. The observation includes a window of length 10, with past observations and actions explicitly stacked. There is no ablation on shorter/longer windows or on relying on the LSTM alone vs. explicit stacking. This is important since history is central to the POMDP formulation.
    - Action space design. The decision space is discretized to a small set of frequencies for each of the four groups. There is no ablation on:
      - restricting or expanding the available frequency choices;
      - tying vs. decoupling the three navigation groups (localization, planning, offboard control).
    - Reward weighting and thresholds:
      - CPU penalty threshold at 95% utilization and weight λ_cpu = 0.1 are design-critical, but there is no sensitivity analysis showing trade-offs with tracking, success rate, or CPU usage.
      - Similarly, λ_traj and λ_est are both 0.01; it would be useful to see whether skewing these weights leads to qualitatively different policies or performance.
    - Collision / termination reward.
      Equation (1) and Section 3.3 describe a terminal collision penalty λ_col that dominates the reward and terminates the episode.

However, the paper does not explicitly state what terminal reward is given on successful mission completion.

  - Baselines could be stronger and more consistent with the proposed action space:
    - The learned policy can independently choose frequencies for flight control and each navigation group.
    - The baselines, however, are limited to strictly increasing, “matched” tuples (100–20–20–20, …, 400–50–50–50) and do not include “mixed” schedules such as high-frequency control with lower-frequency planning or vice versa.
    - Since one motivation of the method is that optimal frequencies differ across modules and contexts, a fairer comparison would include hand-designed “mixed” schedules representative of expert tuning.
    - Another informative experiment would be to constrain the learned policy to the same action manifold as the baselines (matched tuples only) to isolate the benefit of adaptivity vs. access to a richer action space.

  - No safety-focused baseline in a collision-heavy setting:
    - No method achieves zero collisions; success rates are < 100% across the board.
    - Since safety is weighted very heavily in the reward, it would be informative to include at least one baseline explicitly tuned for maximum safety (e.g., maximum navigation frequencies with conservative planning parameters) and show the trade-off between zero-collision operation and other metrics (CPU, trajectory deviation, flight duration).

  - Limited qualitative insight into the learned policy:
    The paper argues that the scheduler adapts to runtime context, but offers little direct evidence:
    - No visualization of correlations between observation variables and chosen actions (e.g., obstacle distance vs. planning frequency, high velocity vs. control frequency).
    - No histograms of action choices to illustrate which combinations of frequencies are considered “good” or “bad” by the policy.
    Such analyses would both validate the interpretation that the policy is context-aware and provide guidance to practitioners.

  - Real-world experiment description is too superficial to add real evidence:
    - Section 5.3 and Fig. 7 essentially show a flying drone and ROS visualization without any quantitative metrics.
    - It is already known from the HIL setup that the stack runs on real hardware.
    - Without statistics such as:
      - mean flight duration,
      - success rate over multiple flights,
      - trajectory deviation (even approximate),
      the real-world experiment is more of a demo than an evaluation, and could be streamlined or replaced by a small quantitative study.
    - The ROS navigation visualization (Fig. 7b) lacks a legend explaining the color coding, which further limits its usefulness.

  - Clarity issues and missing details:
    - Construction of the policy input. While Section 3.1 explains that observations are provided as a fixed-length sliding window with actions included, it remains unclear how the sliding-window observations are fed into the network:
      - Are all stacked observations simply concatenated and flattened before the LSTM?
      - Is there any separate preprocessing per feature type (system, schedule, flight, environment) before stacking?
      - How is the previous action incorporated (one-hot, numeric frequencies, or both)?
    - PPO hyperparameters: The paper does not specify key PPO training settings: learning rates, clipping parameter, entropy coefficients, batch sizes, rollout length, etc. This hinders reproducibility and makes it difficult to interpret the training curves.
    - Figure 5b (episodic return): The figure caption and discussion say “episodic return initially decreases and subsequently stabilizes,” but the plotted curve appears to start relatively high before dropping and stabilizing. Some explanation of the early behavior (e.g., initialization, differing episode lengths, or reward scale changes) would be useful.
    - Figure 5g (“loss”): It is unclear what “loss” refers to here and how it relates to the individual losses in Fig. 5d–5f. Please clarify.
    - Impact statement: The impact statement is missing.

---

> ### Author Rebuttal · Authors · 2026-03-31
>
> # Question 1: Observation encoding and history
> We thank the reviewer for this question. Our current implementation stacks all observation variables together with past scheduling actions, over the last 10 decision steps and feeds the resulting sequence to an LSTM-based RPPO policy without separate temporal preprocessing. The history length of 10 was selected as a practical compromise rather than a tuned hyperparameter. We will include a sensitivity analysis of history length in the revision.
> # Question 2: Action space and baselines
> We thank the reviewer for raising this point. The current discrete frequency space already contains 500 static combinations, making exhaustive evaluation impractical, so we reported representative cases only. We will further include mixed schedules with decoupled control and navigation frequencies here and in the revision. These additional static combinations show markedly higher tracking deviation (Fig. A) and lower success rates (Fig. B) than the baselines in the manuscript, indicating that the reported baselines are already relatively strong. Restricting the learned policy to the same static manifold would be less informative, because its key benefit is online adaptation to runtime conditions.
>
> Fig. A: https://i.postimg.cc/vmMsHtTX/Deviation.png
>
> Fig. B: https://i.postimg.cc/pdxHXJrq/Success-Rate.png
>
> # Question 3: Reward design and sensitivity
> Thank you for this important question. We use a penalty-only reward. Since episodes end at collision or mission completion and collision incurs a large terminal penalty, no separate positive success reward is needed. The coefficients were set according to the role of each term rather than precise tuning: tracking and localization errors are dense penalties with equal small weights, CPU overload is a soft feasibility boundary with a medium weight, and collision is terminal and safety-critical with the largest weight. We will include a sensitivity analysis covering each λ in the revision.
> # Question 4: Learned policy behavior
> We thank the reviewer for pointing this out. We will add a 30 s in-flight visualization of obstacle distance, vehicle speed, and the selected localization, planning, and waypoint frequencies to improve interpretability (Fig. C). It shows that the policy favors higher localization and offboard-control frequencies in fast, obstacle-sparse flight, but shifts toward higher planning frequency when obstacles become closer to improve avoidance responsiveness.
>
> Fig. C: https://i.postimg.cc/HsKM7R0c/Policy-Behavior.png
>
> # Question 5: Training details and losses
> We thank the reviewer for highlighting this issue. We will add the following hyperparameter table (FIg. D) in the revision. Full details (training configuration, hyperparameters and automation scripts) can be found in the code link provided in the paper.
>
> Fig. D: https://i.postimg.cc/MGw4RFdx/Hyperparameters.png
>
> The “loss” in Fig. 5g is the total PPO training objective used for optimization rather than an isolated metric, and it includes the policy, value, and entropy terms with their coefficients. For Fig. 5b, the high value at the beginning mainly reflects small-sample noise in the earliest completed episodes rather than good initial performance, and under our penalty-based reward, broader early exploration can temporarily drive the return downward before training becomes more stable.
> # Question 6.1: Evaluation tasks
> We appreciate this important question. All 50 episodes use the same map, start-goal specification, mission setting, and navigation configuration, isolating scheduling as much as possible. The remaining variation arises mainly from inherent HIL and navigation stochasticity, including execution-time variability on real embedded hardware, OS-level jitter, and randomness in sensing and navigation. The task is still nontrivial, with a straight-line distance of about 50 m and 8 obstacles along the route.
> # Question 6.2: Real-world experiment
> We thank the reviewer for pointing this out. Compared with HIL, scaling real-flight experiments is difficult in our setup due to manual and time-consuming vehicle preparation and reset, limited battery life, and collision risk. We therefore report a preliminary real-flight comparison in a sparse forest (Fig. E) with one trial per method, where the learned policy shows relatively low trajectory error (Fig. F) and more effective CPU usage (Fig. G), consistent with the manuscript. We will add further real-flight experiments in the revision.
>
> Fig. E: https://i.postimg.cc/28W48qzj/Forest.jpg
>
> Fig. F: https://i.postimg.cc/Y0sbcTfY/Mean-Trajectory-Error.png
>
> Fig. G: https://i.postimg.cc/4ySBCjbt/Mean-CPU-Usage.png
>
> # Ethical Concerns
> The prompts are injected by ICML to identify LLM reviewing.

---

> > ### Author Rebuttal · Reviewer_KVq4 · 2026-04-03
> >
> > Thank you for the extensive clarifications. Concerns have been mostly addressed.
> > - Can you evaluate the method on other Gazebo worlds from Appendix E (Village, City, ...)?
> > - The impact statement is missing

---

> > > ### Author Response · Authors · 2026-04-05
> > >
> > > # Q1: Other Worlds/scenarios
> > >
> > > We appreciate the reviewer’s question. We evaluated the policy trained in the Pillars world in 4 additional unseen Gazebo worlds, namely Village, City, BigCity and Forest, without any transfer, fine-tuning, or retraining. Each method was evaluated for 50 episodes in each scenario.
> > >
> > > The learned policy shows good cross-world generalization. In Village, it achieves the highest success rate of 82% (Fig. F). In City, it achieves the highest success rate of 70% (Fig. G) while maintaining relatively low trajectory deviation. In BigCity, it again attains the highest success rate of 68% (Fig. H), with deviation remaining competitive. In Forest, it reaches a tied-best success rate of 86% (Fig. I). Averaged over the four unseen worlds, the learned policy achieves the highest success rate at about 76.5%, compared with about 73% for the best fixed baseline, while keeping deviation in a consistently low range. These results support the policy's generalization ability across diverse unseen scenarios without transfer or retraining.
> > >
> > > Fig. F: https://i.postimg.cc/fWGcV1MW/Village.png
> > >
> > > Fig. G: https://i.postimg.cc/Njd3hFCy/City.png
> > >
> > > Fig. H: https://i.postimg.cc/bY7xZ4qp/Big-City.png
> > >
> > > Fig. I: https://i.postimg.cc/ryvWJ986/Forest.png
> > >
> > > # Q2: Impact Statement
> > >
> > > This work addresses a important problem in resource-constrained UAV autonomy: enabling fully onboard scheduling of a tightly coupled autopilot pipeline, where localization, planning and flight control are integrated on a single computing platform. The work advances understanding and practice by showing that scheduling in such unified architectures can be framed as a partially observable sequential decision problem, and that reinforcement learning can be used to adapt execution rates based on runtime feedback. The proposed framework combines a unified system architecture, a POMDP-based formulation, and a policy trained with hardware-in-the-loop execution, providing a concrete example of how learning-based approaches can optimize complex embedded robotic systems under delayed and cross-coupled interactions among modules.
> > >
> > > The contributions of this work are relevant to resource-constrained embedded systems and embodied AI platforms that require coordinated scheduling of multiple tightly coupled workloads. In particular, the framework provides system-level support for autonomous flight, facilitating higher mission success rate and more stable trajectory tracking under compute limits. In this sense, the scope of impact is specialized but meaningful, and the ideas may inform scheduling and runtime adaptation in other systems facing similar resource constraints.
> > >
> > > This work also has limitations. The current policy focuses on frequency-level task scheduling and does not address kernel-level scheduling, energy-aware management, or other low-level system controls. Future work could extend these capabilities to provide more comprehensive system optimization.
> > >
> > > The societal impact of this work is mixed. On the positive side, this work may help broaden the practical deployment of autonomous UAV systems under strict size, weight, and power constraints. By improving how limited onboard computation is allocated across localization, planning, and flight control, the proposed framework could support more capable operation in applications such as infrastructure inspection, environmental monitoring, precision agriculture, emergency response. More broadly, it points toward a future in which smaller and lower-cost UAV platforms can perform increasingly complex autonomous tasks without relying on heavier or more fragmented computing architectures.
> > >
> > > On the negative side, the same capability could be applied in high-risk or harmful contexts, and failures of a learned scheduler could also increase the risk of unsafe behavior if deployed without sufficient safeguards. Therefore, deployment in safety-critical settings should require conservative safety constraints, runtime monitoring, fail-safe mechanisms, and appropriate human oversight.

---

### Official Review · Reviewer_V9JB · 2026-03-10

**Soundness:** 2
**Presentation:** 2
**Significance:** 3
**Originality:** 3
**Overall Recommendation:** 5
**Confidence:** 4

**Summary:**

The manuscript presents UAV2, a unified and adaptive scheduling framework for UAV autopilot systems. By integrating navigation and flight‑control onto a single onboard computer, the authors cast the scheduling problem as a POMDP and train a PPO agent to select discrete execution frequencies for four task groups. The reward penalises trajectory deviation from the planned and ground‑truth paths, high CPU utilization, and collisions, encouraging a balance between tracking accuracy, computational feasibility, and safety. Experiments are performed in a HIL simulation and a limited real‑world flight test. The learned policy achieves an 86 % success rate---substantially higher than four fixed‑rate baselines---while keeping CPU usage well below saturation and reducing tracking error variance.

**Compliance With Llm Reviewing Policy:**

Affirmed.

**Final Justification:**

After carefully weighing the original review together with the authors' rebuttal, I conclude that the paper now meets the standard for acceptance. The authors have substantially strengthened the manuscript by addressing every major concern I raised, and the additional experimental evidence convinces me that the contribution is both technically sound and practically significant.

**Key Questions For Authors:**

- How does performance vary when the lambdas are altered? An ablation would show whether the observed gains depend on careful hand‑tuning.
- What specific advantages (e.g., existing codebase, library compatibility) led you to retain ROS 1 and Gazebo Classic despite their deprecation? Could you comment on whether a migration to ROS 2/Ignition would be feasible?
- Have you tried UAV2 on a different SoC (e.g., Cortex‑A53) or with a reduced core count? Evidence of the scheduler's adaptability to varied compute budgets would strengthen the claim of generality.
- Can you provide examples of episodes where the learned policy crashes, exceeds CPU limits, or leads to collisions? Understanding these failure modes is crucial for safety‑critical deployment.
- In Section 5.3, a real flight experiment is mentioned, but quantitative results are absent. Could you report success rate, trajectory error, and CPU usage from a physical flight to complement the HIL results?

**Limitations:**

yes

**Strengths And Weaknesses:**

Modeling scheduling as a POMDP is well justified, as scheduling actions have delayed, cross‑coupled effects on state estimation and control that are hard to capture with analytic models. The use of a large collision penalty ensures safety is treated almost as a hard constraint, and the CPU‑overload penalty provides a soft feasibility bound. All baselines share the same map, start/goal positions, and 50 independent episodes; four complementary metrics are reported, giving a clear picture of trade‑offs. The adaptive policy consistently outperforms static‑rate baselines in success rate and exhibits lower tracking error variance, supporting the claim that RL can learn effective scheduling policies. However, failure cases are not discussed; seeing episodes where the policy collapses or violates real‑time constraints would strengthen confidence in safety. Moreover, the safety argumentation is not sound compared to modern safety-by-design AI development methodologies. Finally, reliance on ROS 1 undermines the software stack's futureproofness.

---

> ### Author Rebuttal · Authors · 2026-03-31
>
> # Weakness 2: Safety Scope and Safeguards
> Thank you for pointing this out. We acknowledge that this work do not provide a formal safety argument. Our contribution is an RL-based adaptive scheduling method for resource-constrained UAV autopilot systems. Considering safety, the policy is restricted to bounded task-group frequency selection, and cannot modify kernel priorities or scheduling classes. Also, the collisions receive the strongest penalty during training and immediately terminate the episode. We will revise the paper to present these mechanisms as engineering safeguards rather than formal guarantees, and point safety-by-design AI for UAVs as an important future work.
> # Question 1: Reward Sensitivity to λ
> We appreciate the reviewer’s question. We will include an ablation covering each λ in the revision to justify our methodology. The current coefficients were chosen according to the role of each reward term. Tracking and localization errors are dense penalties incurred throughout flight and therefore use equal small weights. CPU overload is defined as a soft feasibility boundary, activated only when utilization exceeds a preset threshold, and therefore uses a medium weight. Collision is terminal and safety-critical, so it receives the largest weight. We expect the gains to arise mainly from the reward structure rather than precise hand-tuning magnitudes.
> # Question 2 & Weakness 3: ROS1 / Gazebo Classic Choice and Migration
> Thank you for this valuable question. We use Gazebo Classic and ROS1 because HIL is required for training, where many existing UAV HIL tools are still built on this stack. For example, PX4 documentation continues to list Gazebo Classic as a supported HIL backend [1]. However, our method is not inherently tied to Gazebo Classic or ROS1. The method can be migrated to ROS2-Ignition based systems through engineering adaptations, such as ROS2-MAVLink interfaces, an Ignition-MAVLink plugin for HIL, and mechanisms for exposing ROS2 tasks as schedulable units. We will clarify this point in the revision and present this migration as an important direction for future work.
>
> [1] PX4 User Guide v1.15
>
> # Question 3: Adaptation to Different SoC
> We thank the reviewer for this question. We have conducted an experiment to evaluate the resulting deviation (Fig. A) and success rate (Fig. B) of the competing methods on different number of CPU cores and frequencies. The results show that zero-shot generalization to unseen hardware settings is limited, where the policy performs worse than the certain baselines on a single underclocked core. This is expected as scheduling is inherently hardware-dependent, and our framework is designed to learn an effective policy for the target platform from runtime feedback. When deployed on a new SoC, the same framework can be retrained or fine-tuned to obtain a strong platform-specific scheduler. We will add this discussion to the revision.
>
> Fig. A: https://i.postimg.cc/7PjMwjR7/Deviation.png
>
> Fig. B: https://i.postimg.cc/CM9HS92k/Success-Rate.png
>
> # Question 4 & Weakness 1: Failure Cases and Safety Analysis
> We thank the reviewer for raising this point. We will include representative failure episodes and their corresponding analysis in the revision. Below is an example of obstacle distance and scheduling decisions when collision happened (Fig. C). In this case, after reducing the planning frequency to allocate more compute budget to the localization module, the policy failed to raise the planning frequency again when another obstacle was getting close. This insufficient response delayed effective avoidance updates, and the UAV ultimately collided with an obstacle and crashed. We will also provide discussions on how such failures may be mitigated in the revision.
>
> Fig. C: https://i.postimg.cc/MGhTHccQ/Nearest-Lio-Ego.png
>
> # Question 5: Quantitative Real-Flight Results
> Thank you for pointing this out. We note that, unlike HIL, scaling up real-flight experiments in our setup is challenging because vehicle preparation and reset require manual and time-consuming effort, battery life is limited, and collisions can damage the UAV. We therefore report a preliminary real-flight comparison between the learned policy and the baselines, conducted in a sparse forest (Fig. D), using one flight trial for each method. The learned policy shows relatively low trajectory error (Fig. E) together with more effective CPU usage (Fig. F), consistent with the results in the manuscript. We will also include additional real-flight experiments in the revision.
>
> Fig. D: https://i.postimg.cc/28W48qzj/Forest.jpg
>
> Fig. E: https://i.postimg.cc/Y0sbcTfY/Mean-Trajectory-Error.png
>
> Fig. F: https://i.postimg.cc/4ySBCjbt/Mean-CPU-Usage.png

---

> > ### Author Rebuttal · Reviewer_V9JB · 2026-04-01
> >
> > The authors have responded constructively to all of my original concerns and have added useful material (ablation study for the reward weights, discussion of ROS 1 vs. ROS 2 migration, experiments on different CPU core counts, failure‑case analysis, and preliminary real‑flight plots). These additions address the major points I raised about safety safeguards, reward‑weight sensitivity, middleware choice, hardware portability, and failure analysis.
> >
> > However, two aspects remain insufficiently resolved for a definitive reassessment of the paper's scores:
> >
> > Reward‑weight ablation:
> > The rebuttal states that an ablation will be added, but does not indicate the outcome. I would like to see quantitative results showing how performance varies when each lambda is changed. In particular, does the policy's success rate remain robust when the collision penalty is reduced, or the CPU weight is increased?
> >
> > Real‑flight evaluation:
> > The authors provide only a single trial per method in the forest experiment. With such a tiny sample size, statistical significance cannot be established. I request at least 3-5 repeated flights per baseline and per learned policy, along with variance measures (e.g., the standard deviation of trajectory error and CPU usage). This will allow me to assess whether the HIL gains translate reliably to real hardware.
> >
> > Follow-up Questions for the Authors
> >
> > Reward‑weight sensitivity:
> > Please present the ablation results you plan to add. Specifically, which lambda contributes most to the observed performance gains, and what is the performance drop when each term is removed, or its weight is significantly altered?
> >
> > Real‑flight statistics:
> > Do you have additional flight trials (or can you run them) to report the mean and standard deviation of the trajectory error and CPU usage for each method? If not, could you at least provide a justification for why a single trial is representative, perhaps by citing repeatability of the HIL environment?
> >
> > Hardware-generalization strategy:
> > In the new SoC experiments, the policy underperforms on a single under‑clocked core. Do you envisage a transfer-learning approach (e.g., fine-tuning) that requires fewer training episodes than training from scratch? If so, could you include a brief experiment demonstrating this?
> >
> > Addressing these points would allow me to evaluate the robustness of the proposed method fully and, if satisfactory, consider raising the overall recommendation.

---

> > > ### Author Response · Authors · 2026-04-06
> > >
> > > # Question 1: Reward Ablation
> > >
> > > We appreciate the careful reading and insightful feedback. We now provide the ablation and sensitivity analysis results completed so far. Current results already support our main qualitative conclusion that strong performance mainly depends on a proper balance between the dense penalties weighted by $\lambda_{traj}$ and $\lambda_{est}$ and the sparse but strong discouragement imposed by $\lambda_{col}$, while $\lambda_{cpu}$ provides supportive resource-aware regularization. A more comprehensive version will be included in the revision.
> > >
> > > We study three reward modifications:
> > >
> > > * Group A scales all weights together while preserving their ratios.
> > > * Group B removes one term at a time.
> > > * Group C keeps the same values but reassigns them across terms.
> > >
> > > |Setting|$\lambda_{traj}$ & $\lambda_{est}$|$\lambda_{cpu}$|$\lambda_{col}$|
> > > |---|---|---|---|
> > > |Origin|0.01|0.1|1|
> > > |||||
> > > |A1 $(\times 0.1)$|0.001|0.01|0.1|
> > > |A2 $(\times 10)$|0.1|1|10|
> > > |||||
> > > |B1 $(\lambda_{traj}$ & $\lambda_{est} = 0)$|0|0.1|1|
> > > |B2 $(\lambda_{cpu} = 0)$|0.01|0|1|
> > > |B3 $(\lambda_{col} = 0)$|0.01|0.1|0|
> > > |||||
> > > |C1|0.01|1|0.1|
> > > |C2|0.1|0.01|1|
> > > |C3|0.1|1|0.01|
> > > |C4|1|0.01|0.1|
> > > |C5|1|0.1|0.01|
> > >
> > > All variants were trained for 1000 episodes and evaluated over 50 test episodes. Results are reported in Fig. A.
> > >
> > > Fig. A: https://i.postimg.cc/NjkQ7xQD/Ablation.png
> > >
> > > Group A indicates that relative weighting is more important than the absolute reward scale. Jointly scaling all weights leads to little performance change, as success rate, deviation, and CPU time all remain similar to the original setting.
> > >
> > > Group B clarifies the role of each term. In B1, removing $\lambda_{traj}$ and $\lambda_{est}$ eliminates the dense step-wise learning signal, and training stays close to random exploration. Success drops from 86% to 42%, and deviation increases from 0.15 to 0.35. This indicates that **the deviation-related penalty is the main source of dense supervision for policy learning**. In B2, removing $\lambda_{cpu}$ makes the policy ignore computational cost and choose overly aggressive frequencies, which raises the CPU time. Such behavior increases resource competition among tightly coupled tasks, leading to CPU overload and scheduling delays. These timing instabilities degrades estimation and planning quality, which increase the deviation and reduce the success rate. In B3, **removing $\lambda_{col}$ creates an undesirable incentive for early collision termination because shorter episodes reduce the accumulated penalties associated with $\lambda_{traj}$, $\lambda_{est}$, and $\lambda_{cpu}$**. Success drops sharply to 28%, and the CPU time is also lower.
> > >
> > > Group C further shows that the critical design choice is the ratio between the deviation-related weights and $\lambda_{col}$. Once this balance is disturbed, success drops significantly in C1–C5. In particular, settings with low $\lambda_{col}$ perform the worst, indicating that **$\lambda_{col}$ needs to be sufficiently larger than the deviation-related penalty accumulated during a successful episode. Otherwise, the policy may converge to a degenerate strategy that exploits early collision termination**.
> > >
> > > # Question 2: Real-World Test
> > >
> > > We additionally report results from 10 real-flight trials per method in a sparse forest environment. The learned policy achieves the highest success rate of 100%, and a competitive low trajectory-tracking deviation (Table and Fig. H). The results directly address the concern about representativeness, and support the sim-to-real ability of our method.
> > >
> > > |Method|Deviation (m)|Std.|CPU (ms)|Std.|Success Rate (%)|
> > > |---|---|---|---|---|---|
> > > |A|0.40|0.10|25.51|1.67|40|
> > > |B|0.24|0.01|39.98|1.11|80|
> > > |C|0.16|0.02|52.68|4.10|90|
> > > |D|0.15|0.01|62.94|1.60|70|
> > > |Policy|0.17|0.01|67.45|2.48|100|
> > >
> > > Fig. H: https://i.postimg.cc/1X8j4yz3/Real-Sparse-Forest.png
> > >
> > > # Question 3: Hardware Transfer Learning
> > >
> > > We conducted a preliminary two-stage transfer experiment. First, we increased the CPU-time penalty from 0.1 to 0.5, temporarily disabled the collision penalty and termination, raised the entropy coefficient to 0.05, reset the step counter, and started training. This stage was designed to promote rapid adaptation to the tighter compute budget on the target hardware. Once CPU overload events became infrequent, we restored the original reward setting and continued training until reconvergence.
> > >
> > > We transferred the policy from 2×Cortex-A76 to a single Cortex-A55. In Fig. K, transfer training reaches a comparable loss regime at about 20k steps, whereas training from scratch requires about 200k steps. In Fig. L, transfer improves success from 46% to 62%, reduces deviation from about 0.42 m to 0.26 m, and slightly lowers CPU time. Overall, these preliminary results provide encouraging evidence for cross-hardware transfer, and we will include them in the revision.
> > >
> > > Fig. K: https://i.postimg.cc/vTYmCz7H/loss-and-transfer-loss.png
> > >
> > > Fig. L: https://i.postimg.cc/v84HF9vn/Transferred.png

---

### Official Review · Reviewer_prN4 · 2026-03-12

**Soundness:** 3
**Presentation:** 3
**Significance:** 3
**Originality:** 3
**Overall Recommendation:** 5
**Confidence:** 3

**Summary:**

The paper studies a unified framework for scheduling in UAV autopilot systems using reinforcement learning. Specifically, in contrast to literature which deploys navigation and flight control separately restricting system-wide observability, the paper proposes a single onboard computing platform and learns the scheduling between the navigation and flight control from runtime execution feedback. Empirical results demonstrate improvement in performance and robustness.

**Compliance With Llm Reviewing Policy:**

Affirmed.

**Final Justification:**

I appreciate the discussion from the authors and adjusted the score.

**Key Questions For Authors:**

1. Can the authors clarify the choice of the particular frequencies while defining the action sets?
2. While the multi-objective reward function is depictive, why are the values of $\lambda_{traj}$ and $\lambda_{est}$ chosen to be significantly low?
3. Pertaining to figure 5, does a steady-state error persist when the number of iterations are increased?

**Limitations:**

Yes.

**Strengths And Weaknesses:**

Strengths:
1. The idea of using a single computing platform is impressive; it naturally alleviates some of the latency issues associated with multi-platform setups.

Weaknesses:
1. Performing training directly in the HIL setup can be costly; I am not sure if training in software simulation (inclusive of all the uncertainties and sensor noise appropriately modeled) would be a more viable option.
2. Some of the hyperparameter values are missing in the experiments, reducing reproducibility.
3. The experimental evaluations seem restrictive to me, since the comparison is against a set of static policies, which do not seem to be benchmark policies; furthermore, the evaluation results seem to be a bit weak (for instance, a collision rate of 24\% in figure 6 seems to be quite high).

---

> ### Author Rebuttal · Authors · 2026-03-31
>
> # Weakness 1: HIL Training Cost
> Thank you for pointing this out. Since the RL policy targets real onboard autopilot stacks, training must be performed with HIL to capture the actual system effects relevant to scheduling, such as CPU utilization and task execution times. Also, because both SIL and HIL require physics simulation and end-to-end execution of the autopilot pipeline, HIL does not significantly increase the overall computational cost relative to SIL.
> # Weakness 2: Missing Hyperparameters
> We thank the reviewer for highlighting this issue. We will add the following hyperparameter table in the revision. Full details (training configuration, hyperparameters and automation scripts) can be found in the code link provided in the paper.
>
> |Item|Value|
> |---|---|
> |Policy|MlpLstmPolicy|
> |Learning rate|3E-04|
> |Batch size|10|
> |n_epochs|10|
> |Gamma|0.99|
> |GAE lambda|0.95|
> |Clip range|0.2|
> |Entropy coefficient|0|
> |Value loss coefficient|0.5|
> |Max grad norm|0.5|
> |Normalize advantage|TRUE|
> |target_kl|None|
> |Optimizer|Adam|
> |Adam betas|(0.9, 0.999)|
> |Adam eps|1E-05|
> |Step rate|10 Hz|
>
> |Component|Value|
> |---|---|
> |Observation space|Box(80,)|
> |Action space|MultiDiscrete([4,5,5,5])|
> |Feature extractor|FlattenExtractor|
> |Recurrent core|LSTM|
> |LSTM layers|1|
> |LSTM hidden size|256|
> |Actor MLP|64→64(Tanh)|
> |Critic MLP|64→64(Tanh)|
> |Action head|Linear(64→19 logits;4,5,5,5)|
>
> # Weakness 3.1: Limited Baselines
> Thank you for the comment. Fixed-rate configurations reflect common practice in many UAV autopilot systems and research pipelines, where module frequencies are selected offline and remain fixed during execution [1-4]. As for heuristic alternatives, designing a broadly effective hand-crafted rule is difficult in our setting because (i) the effects of a scheduling decision are only partially observable at runtime and (ii) scheduling actions often affect the system through cross-coupled interactions among localization, planning, and control [5,6]. This is also the core motivation for constructing a POMDP formulation and RL method to schedule UAV tasks.
> # Weakness 3.2: Weak Performance
> We thank the reviewer for raising this point. We would like to clarify that the collision rate of the learned policy in Fig. 6b is 14%, not 24%, as its reported success rate is 86.0%, which consistently outperforms all fixed-rate baselines under identical system configurations. As shown in Fig. 6b, the best baseline achieves a success rate of 69.4%, while our method yields an improvement of +16.6%. We believe these gains provide clear evidence that our method is effective in improving flight performance.
> # Question 1: Action Frequency Choice
> We thank the reviewer for this helpful question. The frequencies were selected from the feasible operating range of each module. Flight control is on the critical path and thus requires higher rates and has shorter execution times, while localization, planning, and offboard control can run at lower rates and are more compute-intensive. We therefore use 100–400 Hz for flight control and 10–50 Hz for navigation-related groups to cover meaningful operating regimes while avoiding unrealistic or persistently overloaded settings [1-4]. This design also keeps the RL problem tractable and avoids unsafe exploration. We will clarify this rationale in the revision.
> # Question 2: Reward Weight Choice
> We appreciate the reviewer’s question. The relatively small values of $\lambda_{traj}$ and $\lambda_{est}$ are intentional, not because tracking or estimation are less important, but because these terms are dense per-step penalties with small numerical ranges and therefore accumulate substantially over an episode. By contrast, the CPU penalty is triggered only beyond the utilization boundary, and the collision penalty is applied only as a terminal event penalty. The coefficient differences therefore reflect the different statistical roles of dense versus sparse reward components. We will include a sensitivity analysis covering each λ in the revision.
> # Question 3: Steady-State Error
> We thank the reviewer for this helpful question. A small nonzero steady-state deviation may persist even with more training. This is expected because the policy is trained in a HIL simulation with realistic sensing noise, stochastic flight perturbations, and OS-level timing variability. Hence, the training signal is inherently noisy, and convergence to a bounded non-zero regime is more realistic than zero error. Fig. 5 shows the smoothed deviation decreases over training, indicating stable learning and scheduling quality improvement in this difficult POMDP problem. We will clarify this interpretation in the revision.
>
> [1] PX4 User Guide v1.15
>
> [2] Meier et al., PX4 framework
>
> [3] Xu et al., FAST-LIO2
>
> [4] Zhou et al., EGO-Planner
>
> [5] Lauri et al., POMDP survey
>
> [6] Salamun et al., weakly-hard real-time survey

---

> > ### Author Rebuttal · Reviewer_prN4 · 2026-04-02
> >
> > I have a follow-up question: Can the authors mention some results on an ablation study which tries to quantify the sensitivity wrt each $\lambda$ coefficient?

---

> > > ### Author Response · Authors · 2026-04-06
> > >
> > > # Ablation Study on λ
> > >
> > > We sincerely thank the reviewer for their time and valuable insights. We now provide the ablation and sensitivity analysis results completed so far. Current results already support our main qualitative conclusion that strong performance mainly depends on a proper balance between the dense penalties weighted by $\lambda_{traj}$ and $\lambda_{est}$ and the sparse but strong discouragement imposed by $\lambda_{col}$, while $\lambda_{cpu}$ provides supportive resource-aware regularization. A more comprehensive version will be included in the revision.
> > >
> > > The study includes three types of reward modifications:
> > > * Group A applies global scaling to all weights while preserving their ratios, which serves to examine whether the policy is relatively insensitive to overall reward scale compared with relative weight proportions.
> > > * Group B removes one reward term at a time to evaluate the contribution of each component to success rate, trajectory quality, and system behavior.
> > > * Group C keeps the same weight values but changes their assignment across terms, which tests whether the original prioritization of collision avoidance, CPU regularization, and trajectory and estimation quality is important for achieving strong performance.
> > >
> > > |Setting|$\lambda_{traj}$ & $\lambda_{est}$|$\lambda_{cpu}$|$\lambda_{col}$|
> > > |---|---|---|---|
> > > |Origin|0.01|0.1|1|
> > > |||||
> > > |A1 $(\times 0.1)$|0.001|0.01|0.1|
> > > |A2 $(\times 10)$|0.1|1|10|
> > > |||||
> > > |B1 $(\lambda_{traj}$ & $\lambda_{est} = 0)$|0|0.1|1|
> > > |B2 $(\lambda_{cpu} = 0)$|0.01|0|1|
> > > |B3 $(\lambda_{col} = 0)$|0.01|0.1|0|
> > > |||||
> > > |C1|0.01|1|0.1|
> > > |C2|0.1|0.01|1|
> > > |C3|0.1|1|0.01|
> > > |C4|1|0.01|0.1|
> > > |C5|1|0.1|0.01|
> > >
> > > All variants were trained for 1000 episodes and evaluated over 50 test episodes. Results are reported in Fig. A.
> > >
> > > Fig. A: https://i.postimg.cc/8C4YFQKc/Ablation.png
> > >
> > > Group A suggests that the relative weighting among $\lambda_{traj}$, $\lambda_{est}$, $\lambda_{cpu}$, and $\lambda_{col}$ matters more than the absolute reward scale. When all weights are scaled together while preserving their ratios, the behavior stays much closer to the original setting than when individual terms are removed or reassigned. Specifically, the success rate remains 84% in A1 and 86% in A2, compared with 86% in the original setting, while the mean deviation stays close at 0.14, and the mean CPU time remains similarly stable around 73 ms.
> > >
> > > Group B clarifies the role of each term. In B1, setting $\lambda_{traj}=\lambda_{est}=0$ removes the dense step-wise learning signal and leaves only the sparse penalties associated with $\lambda_{cpu}$ and $\lambda_{col}$, causing training to remain close to random exploration. This is reflected by a sharp drop in success rate from 86% to 42%, together with an increase in mean deviation from 0.15 to 0.35. This suggests that **the deviation-related penalty is the main source of dense supervision for policy learning**. In B2, setting $\lambda_{cpu}=0$ makes the policy ignore computational cost and prefer overly aggressive frequency selections, which raises the mean CPU time from 73.99 ms to 83.15 ms. Such behavior increases resource competition among tightly coupled tasks, leading to CPU overload and scheduling delays. These timing instabilities degrade estimation and planning quality, which increase the mean deviation to 0.21 and reduce the success rate to 66%. In B3, **setting $\lambda_{col}=0$ creates an undesirable incentive for early collision termination, since shorter episodes reduce the accumulated penalties associated with $\lambda_{traj}$, $\lambda_{est}$, and $\lambda_{cpu}$**. Consistently, B3 yields only 28% success, with mean deviation still elevated at 0.25 but mean CPU time dropping sharply to 35.94 ms.
> > >
> > > Group C further shows that the critical design choice is the ratio between the deviation-related weights $\lambda_{traj}$, $\lambda_{est}$ and the collision weight $\lambda_{col}$. Once this balance is disrupted, success drops to 54%, 50%, 30%, 32%, and 26% in C1-C5, while mean deviation remains in the 0.24-0.26 range, consistently much higher than the original 0.15. In particular, settings with relatively low $\lambda_{col}$, such as C3 and C5, perform especially poorly, supporting the view that **$\lambda_{col}$ needs to be sufficiently larger than the deviation-related penalty that can accumulate during a successful episode. Otherwise, the policy may converge to a degenerate strategy that exploits early collision termination**. By contrast, $\lambda_{cpu}$ mainly acts as a resource-aware soft constraint and is not the dominant factor in the overall optimization direction.

---

### Decision · Program_Chairs · 2026-04-30

**Decision:**

Accept (regular)

**Comment:**

Reviewers raise concerns regarding lack of algorithmic and theoretical novelty, insufficient mathematical clarity of the POMDP formulation and the policy, as well as loss functions. There were also questions regarding sensitivity to lambda and choice of tasks and scenarios, both real-world and simulation. The authors provide significant updates in the rebuttal, which the reviewers like. The final version should include these updates, as well as writing improvements to make the paper more easily readable.